# FedRC: Tackling Diverse Distribution Shifts Challenge in Federated Learning by Robust Clustering

## Abstract

Federated Learning (FL) is a machine learning paradigm that safeguards privacy by retaining client data on edge devices. However, optimizing FL in practice can be challenging due to the diverse and heterogeneous nature of the learning system. Though recent research has focused on improving the optimization of FL when distribution shifts occur among clients, ensuring global performance when multiple types of distribution shifts occur simultaneously among clients—such as feature distribution shift, label distribution shift, and concept shift—remain under-explored.

In this paper, we identify the learning challenges posed by the simultaneous occurrence of diverse distribution shifts and propose a clustering principle to overcome these challenges. Through our research, we find that existing methods fail to address the clustering principle. Therefore, we propose a novel clustering algorithm framework, dubbed as FedRC, which adheres to our proposed clustering principle by incorporating a bi-level optimization problem and a novel objective function. Extensive experiments demonstrate that FedRC significantly outperforms other SOTA cluster-based FL methods. Our code will be publicly available.

## 1 Introduction

Federated Learning (FL) is an emerging privacy-preserving distributed machine learning paradigm. The model is transmitted to the clients by the server, and when the clients have completed local training, the parameter updates are sent back to the server for integration. Clients are not required to provide local raw data during this procedure, maintaining their privacy. However, the non-IID nature of clients' local distribution hinders the performance of FL algorithms (McMahan et al., 2016; Li et al., 2020; Karimireddy et al., 2020b; Li et al., 2021), and the distribution shifts among clients become a main challenge in FL.

**Distribution shifts in FL.** As identified in the seminal surveys (Kairouz et al., 2021; Moreno-Torres et al., 2012; Lu et al., 2018), there are three types of distribution shifts across clients that bottleneck the deployment of FL (see Figure 5):

- *Concept shift*: For tasks of using feature $\mathbf{x}$ to predict label $y$, the conditional distributions of labels $\mathcal{P}(y|\mathbf{x})$ may differ across clients, even if the marginal distributions of labels $\mathcal{P}(y)$ and features $\mathcal{P}(\mathbf{x})$ are shared [1].
- *Label distribution shift*: The marginal distributions of labels $\mathcal{P}(y)$ may vary across clients, even if the conditional distribution of features $\mathcal{P}(\mathbf{x}|y)$ is the same.
- *Feature distribution shift*: The marginal distribution of features $\mathcal{P}(\mathbf{x})$ may differ across clients, even if the conditional distribution of labels $\mathcal{P}(y|\mathbf{x})$ is shared.

**New challenges posed by the simultaneous occurrence of multiple types of distribution shifts.** Despite the success of existing methods in addressing data heterogeneity in FL, challenges arise due to the simultaneous occurrence of multiple types of distribution shifts.

- As depicted in Figure 1(a), existing works that address label distribution shift (Li et al., 2020; Wang et al., 2020b; Karimireddy et al., 2020b), and feature distribution shift (Peng et al., 2019; Wang et al., 2022a; Shen et al., 2021), by training single global models, suffer from a significant performance drop when dealing with concept shifts (Ke et al., 2022).

---

[1] More discussions about the definition of concept shifts can be found in Appendix J.

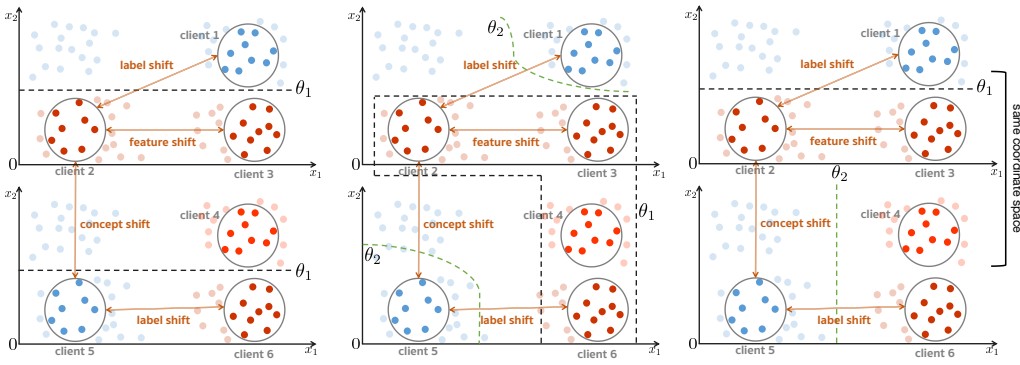

(a) Existing Single-Model Methods (b) Existing Multi-Model Methods (c) Our Method

Figure 1: **Illustration of our *principles of robust clustering*.** Each circle represents a client, with points (features) of varying colors indicating distinct labels. Label shifts are represented by clients exhibiting data points of varying colors, as seen in clients 1 and 2. Feature shifts are exemplified by clients maintaining data points with the same color but having substantial distances between them, as observed in clients 2 and 3. Concept shifts occur when data points at the same position have different labels, as evident in clients 2 and 5. Dashed lines in different colors depict decision boundaries for classifiers of different clusters, i.e., $\boldsymbol{\theta}_1$ and $\boldsymbol{\theta}_2$. *Figure 1(a)* demonstrates that single-model methods are inadequate for handling concept shifts. *Figure 1(b)* shows that current multi-model methods tend to overfit local distributions and can not handle unseen data, like the data points in the top-left corner of Figure 1(b). *Our method (Figure 1(c))* improves model generalization by grouping clients with concept shifts into distinct clusters, while ensuring that clients with only feature or label shifts are placed in the same clusters.

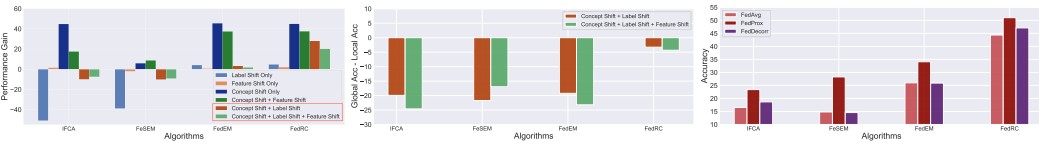

(a) Performance gain over FedAvg    (b) Performance gap (global–local)    (c) Clustered FL + FedProx/FedDecorr

Figure 2: **Performance degradation of existing clustered FL methods.** Figure 2(a) presents the global performance improvements of these methods and our FedRC compared to FedAvg. Figure 2(b) presents the local-global performance gap of these algorithms. Figure 2(c) illustrates the performance of clustered FL when naively combined with single-model methods, such as FedProx (Li et al., 2020) and FedDecorr (Shi et al., 2022). The global distributions are label- and feature-balanced for each concept.

- As shown in Figure 1(b), existing multi-model approaches, such as clustered FL methods (Sattler et al., 2020b; Long et al., 2023; Ghosh et al., 2020), cannot distinguish between different shift types and tend to group data with the same labels into the same clusters, thereby tending to overfit local distributions. As a result, current clustering methods train models with limited generalization abilities, leading to a significant gap between local and global performance (Figure 2(b)).

**Divide-and-Conquer as a solution.** To address the mentioned challenges, we first analyze various distribution shifts and determine which ones can use the same classifier (i.e., decision boundary) and which cannot. In detail, a shared decision boundary can be found for clients without concept shifts, even if they have feature or label shifts:

- When concept shifts occur, the same $x$ can yield distinct $y$, leading to altered decision boundaries.
- While the conditional distributions $\mathcal{P}(y|x)$ remain constant when feature and label shifts happen, a common decision boundary can be determined for these clients.

Therefore, to attain strong generalization capabilities in the face of concept shifts, we employ clustering methods to distinguish concept shifts from other types of shifts. We then train shared decision boundaries for clients that do not have concept shifts. In detail, we leverage the *principles of robust clustering*, as illustrated in Figure 1(c) and text below.

> *separating clients with concept shifts into different clusters,*
> *while keeping clients without concept shifts in the same cluster* [2].

---

[2] A trade-off exists between personalization and generalization, as noted by previous studies (Wu et al., 2022). Our approach prioritizes learning shared decision boundaries, thereby enhancing generalization but potentially

The primary objective of this paper is to identify a clustering method that adheres to the *principles of robust clustering* that existing methods fail: most existing methods cannot distinguish different types of shifts, especially for label and concept shifts, resulting in limited performance gain (Figure 2(a)). Once this objective is achieved, the existing treatments for feature and label shifts can be effortlessly integrated as a plugin to further improve the performance (see Figure 2(c)). To this end, we propose RobustCluster: it allows a principled objective function that could effectively distinguish concept shifts from other shifts. Upon achieving this, we can address the *principles of robust clustering* by developing global models that train clients with the same concepts together. We further extend RobustCluster to the FL scenario and introduce FedRC.

**Our key contributions are summarized as follows:**

- We identify the new challenges posed by the simultaneous occurrence of multiple types of distribution shifts in FL and suggest addressing them using the *principles of robust clustering*. To the best of our knowledge, we are the first to evaluate the clustering results of existing clustered FL methods, such as FeSEM (Long et al., 2023), IFCA (Ghosh et al., 2020), FedEM (Marfoq et al., 2021), and FedSoft (Ruan & Joe-Wong, 2022), under various types of distribution shifts.
- We develop FedRC, a novel soft-clustering-based algorithm framework, to tackle the *principles of robust clustering*. Extensive empirical results on multiple datasets (FashionMNIST, CIFAR10, CIFAR100, and Tiny-ImageNet) and neural architectures (CNN, MobileNetV2, and ResNet18) demonstrate FedRC's superiority over SOTA clustered FL algorithms.
- The FedRC framework enables flexible and effective extensions, including but not limited to, personalization, enhanced interpretability, integration of privacy-preserving techniques, and adaptability to the number of clusters.

## 2   RELATED WORKS

**Federated Learning with distribution shifts.**   As the de facto FL algorithm, (McMahan et al., 2016; Lin et al., 2020) propose using local SGD to reduce communication bottlenecks, but non-IID data distribution among clients hinders performance (Li et al., 2018; Wang et al., 2020b; Karimireddy et al., 2020b;a; Guo et al., 2021; Jiang & Lin, 2023). Addressing distribution shifts is crucial in FL, with most existing works focusing on label distribution shifts through techniques like training robust global models (Li et al., 2018; 2021) or variance reduction methods (Karimireddy et al., 2020b;a). Another research direction involves feature distribution shifts in FL, primarily concentrating on domain generalization to train models that can generalize to unseen feature distributions (Peng et al., 2019; Wang et al., 2022a; Shen et al., 2021; Sun et al., 2022; Gan et al., 2021). These methods aim to train a single robust model, which is insufficient for addressing diverse distribution shift challenges, as decision boundaries change with concept shifts. Concept shift has not been extensively explored in FL, but some special cases have recently emerged as topics of interest (Ke et al., 2022; Fang & Ye, 2022; Xu et al., 2022). (Jothimurugesan et al., 2022) investigated the concept shift by assuming clients do not have concept shifts at the beginning of training. However, in practice, ensuring the absence of concept shifts when local distribution information is unavailable is challenging. In this work, we consider a more realistic but challenging scenario where clients could have all three kinds of shifts with each other, even during the initial training phase.[3]

**Clustered Federated Learning.**   Clustered Federated Learning (clustered FL) is a technique that groups clients into clusters based on their local data distribution to address the distribution shift problem. Various methods have been proposed for clustering clients, including using local loss values (Ghosh et al., 2020), communication time/local calculation time (Wang et al., 2022b), fuzzy $c$-Means (Stallmann & Wilbik, 2022), and hierarchical clustering (Zhao et al., 2020; Briggs et al., 2020; Sattler et al., 2020a). Some approaches, such as FedSoft (Ruan & Joe-Wong, 2022), combine clustered FL with personalized FL by allowing clients to contribute to multiple clusters. Other methods (Long et al., 2023; Marfoq et al., 2021; Zhu et al., 2023; Wu et al., 2023) employ the Expectation-Maximization approach to maximize log-likelihood functions or joint distributions. However, these methods targeting on improving the local performance, therefore overlooked the global performance. In contrast, FedRC shows benefit on obtaining robust global models that perform well on global distributions for each concept, especially when multiple types of shifts occur simultaneously.

---

reducing personalization. To overcome this trade-off, we recommend integrating FedRC with other PFL methods, as detailed in Tables 1 and 4 of our paper.

[3]For more detailed discussions, see Appendix D.

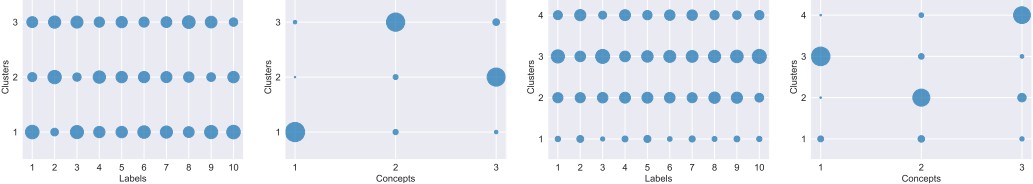

Figure 3: **Clustering results w.r.t. classes/feature styles/concepts.** After data construction, each data point **x** will have a class $y$, feature style $f$, and concept $c$. We report the percentage of data points associated with a class, feature style, or concept assigned to cluster k. For example, for a circle centered at position $(y, k)$, a larger circle size signifies that more data points with class $y$ are assigned to cluster $k$. For feature styles, we only represent $f \in [1, 10]$ here for clearer representation, and the full version can be found in Figure 12 of Appendix I. By the *principles of robust clustering*, we require a clustering method in which clients with the same concept are assigned to the same cluster (for example, Figure 4(b)).

(a) $K = 3$, w.r.t. classes   (b) $K = 3$, w.r.t. concepts   (c) $K = 4$, w.r.t. classes   (d) $K = 4$, w.r.t. concepts

Figure 4: **Clustering results of FedRC w.r.t. classes/concepts.** The number of clusters is selected within the range of $[3, 4]$, while keeping the remaining settings consistent with those used in Figure 3. Due to page limitations, we present the clustering results w.r.t. feature styles in Appendix I.3, Figure 11.

## 3   REVISITING CLUSTERED FL: A DIVERSE DISTRIBUTION SHIFTS PERSPECTIVE

This section examines the results of existing clustered FL methods from the view of distribution shifts.

**Algorithms and experiment settings.**   We employed clustered FL methods, including IFCA (Ghosh et al., 2020), FeSEM (Long et al., 2023), FedEM (Marfoq et al., 2021), and FedSoft (Ruan & Joe-Wong, 2022). We create a scenario incorporating all three types of shifts:
- **Label distribution shift:** We employ LDA (Yoshida et al., 2019; Hsu et al., 2019) with $\alpha = 1.0$, and split CIFAR10 to 100 clients.
- **Feature distribution shift:** Each client will randomly select one feature style from 20 available styles following the approach used in CIFAR10-C (Hendrycks & Dietterich, 2019).
- **Concept shift:** We create concept shifts by setting different $\mathbf{x} \rightarrow y$ mappings following the approach utilized in prior studies (Jothimurugesan et al., 2022; Ke et al., 2022; Canonaco et al., 2021). Each client randomly selects one $\mathbf{x} \rightarrow y$ mapping from the three available $\mathbf{x} \rightarrow y$ mappings.

Upon construction, each client's samples **x** will possess a class label $y$, a feature style $f$, and a concept $c$. We then run the aforementioned four clustered FL algorithms until convergence. For every class $y \in [1, 10]$, feature style $f \in [1, 20]$, or concept $c \in [1, 3]$, we display the percentage of data associated with that class, feature style, or concept in each cluster.

**Existing clustered FL methods fail to achieve the *principles of robust clustering*.**   We present the clustering results of clustered FL methods regarding classes, feature styles, and concepts in Figure 3. Results show that: (1) All existing clustered FL methods are unable to effectively separate data with

concept shifts into distinct clusters, as indicated in the row of Concept; (2) FeSEM, FedEM, and IFCA are not robust to label distribution shifts, meaning that data with the same class labels are likely to be grouped into the same clusters, as shown in the row of Class Labels; (3) FeSEM and IFCA are not resilient to feature distribution shifts, as shown in the row of Feature Styles; (4) FedSoft is unable to cluster clients via the local distribution shifts, due to the ambiguous common pattern of clients in the same cluster evidenced in the FedSoft column.

## 4 OUR APPROACH: FEDRC

Section 3 identified sub-optimal clustering in existing clustered FL methods for the *principles of robust clustering*. To address this, we introduce a new soft clustering algorithm called FedRC in this section. We first present the centralized version (RobustCluster) and then adapt it for FL scenarios as FedRC.

### 4.1 ROBUSTCLUSTER: TRAINING ROBUST GLOBAL MODELS FOR EACH CONCEPT

In this section, we formulate the clustering problem as a bi-level optimization problem and introduce a new objective function to achieve the *principles of robust clustering*.

**Clustering via bi-level optimization.** Given $M$ data sources $\{\mathcal{D}_1, \cdots, \mathcal{D}_M\}$ and the number of clusters $K$, clustering algorithms can be formulated as optimization problems that maximize objective function $\mathcal{L}(\mathbf{\Theta}, \mathbf{\Omega})$ involving two parameters:
- $\mathbf{\Theta} := [\boldsymbol{\theta}_1, \cdots, \boldsymbol{\theta}_K]$, where the distribution for cluster $k$ is parameterized by $\boldsymbol{\theta}_k$. In most cases, $\boldsymbol{\theta}_k$ is learned by a deep neural network by maximizing the likelihood function $\mathcal{P}(\mathbf{x}, y; \boldsymbol{\theta}_k)$ for any data $(\mathbf{x}, y)$ belonging to cluster $k$ (Long et al., 2023; Marfoq et al., 2021) [4].
- $\mathbf{\Omega} := [\omega_{1;1}, \cdots, \omega_{1;K}, \cdots, \omega_{M;K}] \in \mathbb{R}^{MK}$, with $\omega_{i;k}$ denoting the clustering weight assigned by $D_i$ to cluster $k$. The choice of $\mathbf{\Omega}$ varies among different clustering methods (Ruan & Joe-Wong, 2022; Ghosh et al., 2020). Additionally, we usually have $\sum_{k=1}^{K} \omega_{i;k} = 1, \forall i$.

**Objective function of RobustCluster.** Taking the *principles of robust clustering* into account, the expected objective function $\mathcal{L}(\mathbf{\Theta}, \mathbf{\Omega})$ must fulfill the following properties:
- **Maximizing likelihood function.** $\mathcal{L}(\mathbf{\Theta}, \mathbf{\Omega})$ should be positively correlated with the likelihood function $\mathcal{P}(\mathbf{x}, y; \boldsymbol{\theta}_k)$, since the primary goal of RobustCluster is to learn $\boldsymbol{\theta}_k$ that accurately represents the distribution of cluster $k$ ;
- **Adhering to principles of clustering.** $\mathcal{L}(\mathbf{\Theta}, \mathbf{\Omega})$ should be small only when *principles of robust clustering* is not satisfied, that is, clients with concept shifts are assigned into the same clusters.

To this end, we design the following objective function:

$$\mathcal{L}(\mathbf{\Theta}, \mathbf{\Omega}) = \frac{1}{N} \sum_{i=1}^{M} \sum_{j=1}^{N_i} \ln \left( \sum_{k=1}^{K} \omega_{i;k} \mathcal{I}(\mathbf{x}_{ij}, y_{ij}, \boldsymbol{\theta}_k) \right) + \sum_{i=1}^{M} \lambda_i \left( \sum_{k=1}^{K} \omega_{i;k} - 1 \right), \quad (1)$$

where $\mathcal{I}(\mathbf{x}, y; \boldsymbol{\theta}_k) = \frac{\mathcal{P}(\mathbf{x}, y; \boldsymbol{\theta}_k)}{\mathcal{P}(\mathbf{x}; \boldsymbol{\theta}_k)\mathcal{P}(y; \boldsymbol{\theta}_k)} = \frac{\mathcal{P}(y|\mathbf{x}; \boldsymbol{\theta}_k)}{\mathcal{P}(y; \boldsymbol{\theta}_k)} = \frac{\mathcal{P}(\mathbf{x}|y; \boldsymbol{\theta}_k)}{\mathcal{P}(\mathbf{x}; \boldsymbol{\theta}_k)}$. Note that $N_i := |\mathcal{D}_i|$, and $N = \sum_{i=1}^{M} N_i$, and $(\mathbf{x}_{i,j}, y_{i,j})$ is the $j$-th data sampled from dataset $D_i$. Furthermore, $\lambda_i$ is a Lagrange Multiplier for $\mathcal{D}_i$ that enforces the constraint $\sum_{k=1}^{K} \omega_{i;k} = 1$.

**Interpretation of the objective function.** Maximizing the $\mathcal{L}(\mathbf{\Theta}, \mathbf{\Omega})$ can achieve the *principles of robust clustering*. It can be verified by assuming that data $(\mathbf{x}, y)$ is assigned to cluster $k$. In detail:
- **Maximizing $\mathcal{L}(\mathbf{\Theta}, \mathbf{\Omega})$ can avoid concept shifts within the same cluster.** If $(\mathbf{x}, y)$ exhibits a concept shift with respect to the distribution of cluster $k$, $\mathcal{P}(y|\mathbf{x}; \boldsymbol{\theta}_k)$ will be small (toy examples can be found in Figure 8). This will lead to a decrease on $\mathcal{L}(\mathbf{\Theta}, \mathbf{\Omega})$, which contradicts our goal of maximizing $\mathcal{L}(\mathbf{\Theta}, \mathbf{\Omega})$.
- **$\mathcal{L}(\mathbf{\Theta}, \mathbf{\Omega})$ can decouple concept shifts with label or feature distribution shifts.** If $(\mathbf{x}, y)$ exhibits a label or feature distribution shift with respect to the distribution of cluster $k$, $\mathcal{P}(y; \boldsymbol{\theta}_k)$ or $\mathcal{P}(\mathbf{x}; \boldsymbol{\theta}_k)$ will be small. Therefore, $\mathcal{L}(\mathbf{\Theta}, \mathbf{\Omega})$ will not decrease significantly as (1) data points without concept shifts generally have larger $\mathcal{P}(y|\mathbf{x}; \boldsymbol{\theta}_k)$ compared to those with concept shifts, and (2) $\mathcal{L}(\mathbf{\Theta}, \mathbf{\Omega})$ is negatively related to both $\mathcal{P}(y; \boldsymbol{\theta}_k)$ and $\mathcal{P}(\mathbf{x}; \boldsymbol{\theta}_k)$.

We also give a more detailed explanation of how RobustCluster achieves the *principles of robust clustering* using toy examples in Figure 8 and Figure 9 of Appendix G.

---

[4] Here, $\mathcal{P}(\mathbf{x}, y; \boldsymbol{\theta}_k)$ represents the probability density of $(\mathbf{x}, y)$ for the distribution parameterized by $\boldsymbol{\theta}_k$.

### 4.2 OPTIMIZATION PROCEDURE OF ROBUSTCLUSTER

This section elaborates on how to optimize the objective function defined in (1) in practice.

**Approximated objective function for practical implementation.** Note that $\mathcal{I}(\mathbf{x}, y; \boldsymbol{\theta}_k)$ cannot be directly evaluated in practice. To simplify the implementation, we choose to calculate $\mathcal{I}(\mathbf{x}, y; \boldsymbol{\theta}_k)$ by $\frac{\mathcal{P}(y|\mathbf{x};\boldsymbol{\theta}_k)}{\mathcal{P}(y;\boldsymbol{\theta}_k)}$, and both $\mathcal{P}(y|\mathbf{x}; \boldsymbol{\theta}_k)$ and $\mathcal{P}(y; \boldsymbol{\theta}_k)$ need to be approximated. Therefore, we introduce $\tilde{\mathcal{I}}(\mathbf{x}, y; \boldsymbol{\theta}_k)$ as an approximation of $\mathcal{I}(\mathbf{x}, y; \boldsymbol{\theta}_k)$, and we elaborate the refined definition of (1) below:

$$\mathcal{L}(\boldsymbol{\Theta}, \boldsymbol{\Omega}) = \frac{1}{N} \sum_{i=1}^{M} \sum_{j=1}^{N_i} \ln \left( \sum_{k=1}^{K} \omega_{i;k} \tilde{\mathcal{I}}(\mathbf{x}_{i,j}, y_{i,j}, \boldsymbol{\theta}_k) \right) + \sum_{i=1}^{M} \lambda_i \left( \sum_{k=1}^{K} \omega_{i;k} - 1 \right), \tag{2}$$

where

$$\tilde{\mathcal{I}}(\mathbf{x}, y; \boldsymbol{\theta}_k) = \frac{\exp(-f(\mathbf{x}, y, \boldsymbol{\theta}_k))}{C_{y,k}} \tag{3}$$

Note that $f(\mathbf{x}, y, \boldsymbol{\theta}_k)$ is the loss function defined by $-\ln \mathcal{P}(y|\mathbf{x}; \boldsymbol{\theta}_k) + C$ for some constant $C$ (Marfoq et al., 2021) [5]. $C_{y,k}$ is the constant that used to approximate $\mathcal{P}(y; \boldsymbol{\theta}_k)$ in practice. The intuition behind using $\tilde{\mathcal{I}}(\mathbf{x}, y; \boldsymbol{\theta}_k)$ as an approximation comes from:

- The fact of $f(\mathbf{x}, y, \boldsymbol{\theta}_k) \propto -\ln \mathcal{P}(y|\mathbf{x}; \boldsymbol{\theta}_k)$. We can approximate $\mathcal{P}(y|\mathbf{x}; \boldsymbol{\theta}_k)$ using $\exp(-f(\mathbf{x}, y, \boldsymbol{\theta}_k))$. This approximation will only change $\mathcal{L}(\boldsymbol{\Theta}, \boldsymbol{\Omega})$ to $\mathcal{L}(\boldsymbol{\Theta}, \boldsymbol{\Omega}) + C$.
- $C_{y,k}$ is calculated by $\frac{1}{N} \sum_{i=1}^{M} \sum_{j=1}^{N_i} \mathbf{1}_{\{y_{i,j}=y\}} \gamma_{i,j;k} / \frac{1}{N} \sum_{i=1}^{M} \sum_{j=1}^{N_i} \gamma_{i,j;k}$ in this paper, where $\gamma_{i,j;k}$ represents the weight of data $(\mathbf{x}_{i,j}, y_{i,j})$ assigned to $\boldsymbol{\theta}_k$ (c.f. Remark 4.1). Thus, $C_{y,k}$ corresponds to the proportion of data pairs labeled as $y$ that choose model $\boldsymbol{\theta}_k$, and can be used to approximate $\mathcal{P}(y; \boldsymbol{\theta}_k)$.

The optimization steps for RobustCluster are obtained by maximizing (2) that alternatively updates $\gamma_{i,j;k}^t$ and $\omega_{i;k}^t$ using (4), and $\boldsymbol{\theta}_k^t$ using (5). The proof details refer to Appendix A.

$$\gamma_{i,j;k}^t = \frac{\omega_{i;k}^{t-1} \tilde{\mathcal{I}}(\mathbf{x}_{i,j}, y_{i,j}, \boldsymbol{\theta}_k^{t-1})}{\sum_{n=1}^{K} \omega_{in}^{t-1} \tilde{\mathcal{I}}(\mathbf{x}_{i,j}, y_{i,j}, \boldsymbol{\theta}_k^{t-1})}, \quad \omega_{i;k}^t = \frac{1}{N_i} \sum_{j=1}^{N_i} \gamma_{i,j;k}^t, \tag{4}$$

$$\boldsymbol{\theta}_k^t = \boldsymbol{\theta}_k^{t-1} - \eta \frac{1}{N} \sum_{i=1}^{M} \sum_{j=1}^{N_i} \gamma_{i,j;k}^t \nabla_{\boldsymbol{\theta}} f_{i,k}(\mathbf{x}_{i,j}, y_{i,j}, \boldsymbol{\theta}_k^{t-1}). \tag{5}$$

**Remark 4.1** (Property of $\gamma_{i,j;k}$). *We can observe from (5) that $\gamma_{i,j;k}$ can serve as the weight of data point $(\mathbf{x}_{i,j}, y_{i,j})$ that contributes to the update of $\boldsymbol{\theta}_k$, and $\omega_{i;k}$ is the average of $\gamma_{i,j;k}$ over all data points $(\mathbf{x}_{i,j}, y_{i,j})$ in $D_i$.*

**Remark 4.2** (Compare with existing bi-level optimization methods). *EM algorithms are also categorized as bi-level optimization algorithms (Nguyen et al., 2020; Marfoq et al., 2021) and share a similar optimization framework with our method. However, the key difference lies in the design of the objective function. Our proposed objective function is distinct, and we leverage the bi-level optimization framework as one of the possible solutions to optimize it. In Figure 9 of Appendix G, we demonstrate that without our designed objective function, the decision boundary will be unclear, resulting in poor classification performance.*

### 4.3 CONVERGENCE OF ROBUSTCLUSTER

In this section, we give the convergence rate of the centralized clustering algorithm RobustCluster.

**Assumption 1** (Smoothness Assumption). *Assume functions $f(\boldsymbol{\theta})$ are $L$-smooth, i.e. $\|\nabla f(\boldsymbol{\theta}_1) - \nabla f(\boldsymbol{\theta}_2)\| \leq L \|\boldsymbol{\theta}_1 - \boldsymbol{\theta}_2\|$.*

**Assumption 2** (Bounded Gradient Assumption). *Assume that the gradient of local objective functions $\nabla f(\boldsymbol{\theta})$ are bounded, i.e. $\mathbb{E}\left[\|\nabla f(\boldsymbol{\theta})\|^2\right] = \frac{1}{N} \sum_{i=1}^{M} \sum_{j=1}^{N_i} \|\nabla f(\mathbf{x}_{i,j}, y_{i,j}, \boldsymbol{\theta})\|^2 \leq \sigma^2$.*

The assumptions of smoothness (Assumption 1) and bounded gradient (Assumption 2) are frequently employed in non-convex optimization problems (Chen et al., 2018; Mertikopoulos et al., 2020). We introduce the bounded gradient assumption since we only assume $f(\boldsymbol{\theta})$ to be $L$-smooth, and $\mathcal{L}(\boldsymbol{\Theta}, \boldsymbol{\Omega})$ may not be a smooth function. We demonstrate that RobustCluster converges under these standard assumptions and attains the typical $O(1/\epsilon)$ convergence rate.

---

[5]The $-\ln \mathcal{P}(y|\mathbf{x}; \boldsymbol{\theta}_k) + C$ formulation can accommodate widely used loss functions such as cross-entropy loss, logistic loss, and mean squared error loss.

---

**Algorithm 1** FedRC Algorithm Framework

---

**Require:** The number of models $K$, initial parameters $\boldsymbol{\Theta}^0 = [\boldsymbol{\theta}_1^0, \cdots, \boldsymbol{\theta}_K^0]$, global learning rate $\eta_g$, initial weights $\omega_{i;k}^0 = \frac{1}{K}$ for any $i, k$, and $\gamma_{i,j;k} = \frac{1}{K}$ for any $i, j, k$, local learning rate $\eta_l$.

**Ensure:** Trained parameters $\boldsymbol{\Theta}^T = [\boldsymbol{\theta}_1^T, \cdots, \boldsymbol{\theta}_K^T]$ and weights $\omega_{i;k}^T$ for any $i, k$.

1: **for** round $t = 1, \ldots, T$ **do**
2:      **Communicate** $\boldsymbol{\Theta}^t$ to the chosen clients.
3:      **for** client $i \in \mathcal{S}^t$ **in parallel do**
4:          Initialize local model $\boldsymbol{\theta}_{i,k}^{t,0} = \boldsymbol{\theta}_k^t, \forall k$.
5:          Update $\gamma_{i,j;k}^t, \omega_{i;k}^t$ by (4), $\forall j, k$.
6:          Obtain $\boldsymbol{\theta}_{i,k}^{t,\mathcal{T}}$ by locally updating $\boldsymbol{\theta}_{i,k}^{t,0}$ for $\mathcal{T}$ local steps using the equation (6), $\forall k$.
7:          **Communicate** $\Delta\boldsymbol{\theta}_{i,k}^t \leftarrow \boldsymbol{\theta}_{i,k}^{t,\mathcal{T}} - \boldsymbol{\theta}_{i,k}^{t,0}, \forall k$.
8:      $\boldsymbol{\theta}_k^{t+1} \leftarrow \boldsymbol{\theta}_k^t + \frac{\eta_g}{\sum_{i \in \mathcal{S}^t} N_i} \sum_{i \in \mathcal{S}^t} N_i \Delta\boldsymbol{\theta}_{i,k}^t, \forall k$.

---

**Theorem 4.3** (Convergence rate of RobustCluster). *Assume $f_{ik}$ satisfy Assumption 1-2, setting $T$ as the number of iterations, and $\eta = \frac{8}{40L+9\sigma^2}$, we have,*

$$\frac{1}{T} \sum_{t=0}^{T-1} \sum_{k=1}^{K} \left\| \nabla_{\boldsymbol{\theta}_k} \mathcal{L}\left(\boldsymbol{\Theta}^t, \boldsymbol{\Omega}^t\right) \right\|^2 \leq \mathcal{O}\left(\frac{(40L+9\sigma^2)\left(\mathcal{L}^\star - \mathcal{L}^0\right)}{4T}\right),$$

*where $\mathcal{L}^\star$ is the upper bound of $\mathcal{L}(\boldsymbol{\Theta}, \boldsymbol{\Omega})$, and $\mathcal{L}^0 = \mathcal{L}(\boldsymbol{\Theta}^0, \boldsymbol{\Omega}^0)$. Proof details refer to Appendix B.*

### 4.4 FEDRC: ADAPTING ROBUSTCLUSTER INTO FL

In Algorithm 1 and Figure 10, we summarize the whole process of FedRC. In detail, at the beginning of round $t$, the server transmits parameters $\boldsymbol{\Theta}^t = [\boldsymbol{\theta}_1^t, \cdots, \boldsymbol{\theta}_K^t]$ to clients. Clients then locally update $\gamma_{i,j;k}^{t+1}$ and $\omega_{i;k}^{t+1}$ using (4) and (5), respectively. Then clients initialize $\boldsymbol{\theta}_{i,k}^{t,0} = \boldsymbol{\theta}_k^t$, and update parameters $\boldsymbol{\theta}_{i,k}^{t,\tau}$ for $\mathcal{T}$ local steps:

$$\boldsymbol{\theta}_{i,k}^{t,\tau} = \boldsymbol{\theta}_{i,k}^{t,\tau-1} - \frac{\eta_l}{N_i} \sum_{j=1}^{N_i} \gamma_{i,j;k}^t \nabla_{\boldsymbol{\theta}} f_{ik}(\mathbf{x}_{i,j}, y_{i,j}, \boldsymbol{\theta}_{i,k}^{t,\tau-1}), \tag{6}$$

where $\tau$ is the current local iteration, and $\eta_l$ is the local learning rate. In our algorithm, the global aggregation step uses the FedAvg method as the default option. However, other global aggregation methods e.g. (Wang et al., 2020a;b) can be implemented if desired. We include more details about the model prediction in Appendix H.2. We also include the discussion about the convergence rate of FedRC in Appendix C.

## 5 NUMERICAL RESULTS

In this section, we show the superior performance of FedRC compared with other FL baselines on FashionMNIST, CIFAR10, CIFAR100, and Tiny-ImageNet datasets on scenarios that all kinds of distribution shifts occur simultaneously[6]. We also show FedRC outperforms other clustered FL methods on datasets with real concept shifts in Appendix I. For more experiments on incorporating privacy-preserving and communication efficiency techniques, automatically decide the number of clusters, and ablation studies on the number of clusters, please refer to Appendix F and I.

### 5.1 EXPERIMENT SETTINGS

We introduce the considered evaluation settings below (and see Figure 5); more details about implementation details, simulation environments, and datasets can be found in Appendix H.5- H.8.

**Evaluation and datasets.** We construct two client types: participating clients and nonparticipating clients (detailed in Figure 5), and report the 1) *local accuracy*: the mean test accuracy of participating clients; 2) *global accuracy (primary metric)*: the mean test accuracy of nonparticipating clients. It assesses whether shared decision boundaries are identified for each concept.

---

[6]Ablation studies on single-type distribution shift scenarios are included in Appendix I.5, we show FedRC achieves comparable performance with other methods on these simplified scenarios.

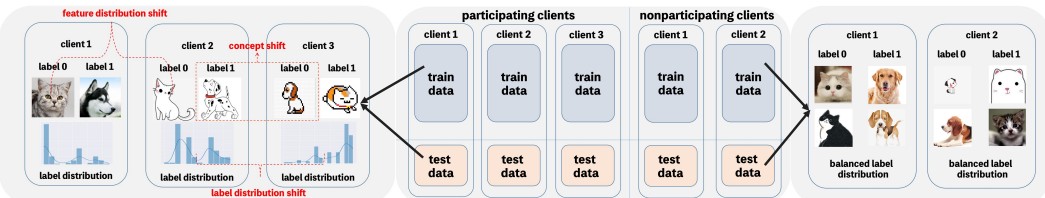

Figure 5: **Illustration of our numerical evaluation protocols.** Clients are divided into two categories: participating clients engage in training, while nonparticipating clients are used for testing. The training and test distributions of each client are identical. Participating clients simulate real-world scenarios and may experience label, feature, and concept shifts. For example, clients 1 and 2 have different label distribution and feature styles (photo or cartoon), while clients 2 and 3 have concept shifts (labels swapped). Nonparticipating clients are utilized to test the robustness of models. Labels on nonparticipating clients are swapped in the same manner as participating clients for each concept.

Table 1: **Performance of algorithms over various datasets and neural architectures.** We evaluated the performance of our algorithms using the FashionMNIST, CIFAR10 and Tiny-ImageNet datasets split into 300 clients. We initialized 3 clusters for clustered FL methods and reported mean local and global test accuracy on the round that achieved the best train accuracy for each algorithm. We report FedRC-FT by fine-tuning FedRC for one local epoch; "CFL (3)" refers to restricting the number of clusters in CFL (Sattler et al., 2020b) to 3. We highlight the best and the second best results for each of the two main blocks, using **bold font** and blue text.

| Algorithm | FashionMNIST (CNN) | | CIFAR10 (MobileNetV2) | | Tiny-ImageNet (MobileNetV2) | |
|---|---|---|---|---|---|---|
| | Local | Global | Local | Global | Local | Global |
| FedAvg | 42.12 ±0.33 | 34.35 ±0.92 | 30.28 ±0.38 | 30.47 ±0.76 | 18.61 ±0.15 | 14.27 ±0.28 |
| IFCA | 47.90 ±0.60 | 31.30 ±2.69 | 43.76 ±0.40 | 26.62 ±3.34 | 22.81 ±0.75 | 12.54 ±1.46 |
| CFL (3) | 41.77 ±0.40 | 33.53 ±0.35 | 41.49 ±0.64 | 29.12 ±0.02 | 23.87 ±1.54 | 11.42 ±2.15 |
| FeSEM | 60.99 ±1.01 | 47.63 ±0.99 | 45.32 ±0.16 | 30.79 ±0.02 | 23.09 ±0.71 | 11.97 ±0.05 |
| FedEM | 56.64 ±2.14 | 28.08 ±0.92 | 51.31 ±0.97 | 43.35 ±2.29 | 28.57 ±1.49 | 17.33 ±2.12 |
| FedRC | **66.51** ±2.39 | **59.00** ±4.91 | **62.74** ±2.37 | **63.83** ±2.26 | **34.47** ±0.01 | **27.79** ±0.97 |
| CFL | 42.47 ±0.02 | 32.37 ±0.09 | 67.14 ±0.87 | 26.19 ±0.26 | 23.61 ±0.89 | 12.47 ±0.46 |
| FedSoft | **91.35** ±0.04 | 19.88 ±0.50 | **83.08** ±0.02 | 22.00 ±0.50 | 70.79 ±0.09 | 2.67 ±0.04 |
| FedRC-FT | 91.02 ±0.26 | **62.37** ±1.09 | 82.81 ±0.90 | **65.33** ±1.80 | **75.10** ±0.24 | **27.67** ±3.13 |

- **Participating clients:** We construct a scenario that encompasses label shift, feature shift, and concept shift issues. For label shift, we adopt the Latent Dirichlet Allocation (LDA) introduced in (Yurochkin et al., 2019; Hsu et al., 2019) with parameter $\alpha = 1.0$. For feature shift, we employ the idea of constructing FashionMNIST-C (Weiss & Tonella, 2022), CIFAR10-C, CIFAR100-C, and ImageNet-C (Hendrycks & Dietterich, 2019). For concept shift, similar to previous works (Jothimurugesan et al., 2022; Ke et al., 2022; Canonaco et al., 2021), we change the labels of partial clients (i.e., from $y$ to $(C - y)$, where $C$ is the number of classes). Unless stated otherwise, we assume only three concepts exist in the learning.
- **Nonparticipating clients:** To assess if the algorithms identify shared decision boundaries for each concept and improve generalization, we create three non-participating clients with balanced label distribution for each dataset (using the test sets provided by each dataset). The labels for these non-participating clients will be swapped in the same manner as the participating clients for each concept.

**Baseline algorithms.** We choose FedAvg (McMahan et al., 2016) as an example in single-model FL. For clustered FL methods, we choose IFCA (Ghosh et al., 2020), CFL (Sattler et al., 2020b), FeSEM (Long et al., 2023), FedEM (Marfoq et al., 2021), and FedSoft (Ruan & Joe-Wong, 2022).

## 5.2 RESULTS OF FEDRC

In this section, we initialize 3 clusters for all clustered FL algorithms, which is the same as the number of concepts.

**Superior performance of FedRC over other strong FL baselines.** The results in Table 1 and Table 2 show that: i) FedRC consistently attains significantly higher global and local accuracy compared to other clustered FL baselines. ii) FedRC achieves a lower global-local performance gap among all the algorithms, indicating the robustness of FedRC on global distribution. iii) The FedRC-FT method, obtained by fine-tuning FedRC for a single local epoch, can achieve local accuracy

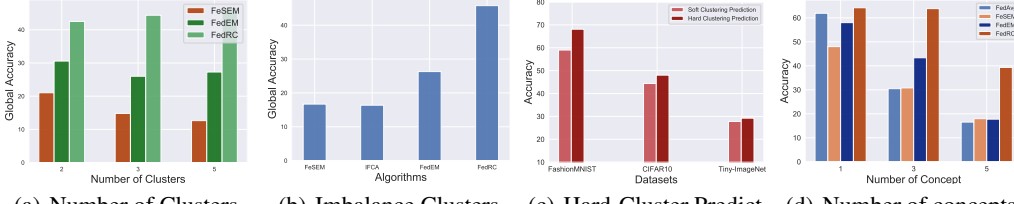

| (a) Number of Clusters | (b) Imbalance Clusters | (c) Hard-Cluster Predict | (d) Number of concepts |

Figure 6: **Ablation study on the number of clusters, the number of concepts, the type of clustering, and the scenarios when the number of samples among clusters is imbalance.** We report the global test accuracy of CIFAR10 with on the round that achieves the best training accuracy. The Figures 6(a), 6(b), and 6(c) use the settings with 100 clients, and obtain the same results as Figure 2. The Figure 6(d) uses the same setting as Table 1. Figure 6(a) shows the performance of clustered FL algorithms with different $K$ values. Figure 6(b) shows the performance of clustered FL methods when the number of samples in each cluster was in the ratio of 8:1:1. Figure 6(c) shows FedRC's performance using soft clustering (ensemble of $K$ clusters) and hard clustering, i.e., utilizing the cluster with the highest clustering weights $\omega_{i;k}$ for prediction. Figure 6(d) shows clustered FL's performance with varying numbers of concepts. We use $\{3, 3, 5\}$ clusters for the scenarios with $\{1, 3, 5\}$ concepts.

comparable to that of clustered FL methods with personalized local models (such as CFL and FedSoft) while maintaining high global performance. Evaluations on more PFL baselines can be found in Table 4 of Appendix I.3.

**FedRC consistently outperforms other methods with varying cluster numbers.** We conducted ablation studies on the number of clusters, as shown in Figure 6(a). The results indicate that FedRC consistently achieves the highest global accuracy across all algorithms.

**The performance improvements of FedRC persist even when there is an imbalance in the number of samples across clusters.** We conducted experiments in which the number of samples in each cluster followed a ratio of 8:1:1, as shown in Figure 6(b). Results show that FedRC maintained a significantly higher global accuracy when compared to other methods.

**The performance improvement of FedRC remains consistent across various concept numbers and in scenarios without concept shifts.** In Figure 6(d), we observe that: 1) FedRC achieves higher global accuracy compared to other methods, and this advantage becomes more significant as the number of

Table 2: **Performance of algorithms on ResNet18.** We evaluated the performance of our algorithms on the CIFAR10 and CIFAR100, which were split into 100 clients.

| Algorithm | CIFAR10 | | CIFAR100 | |
|---|---|---|---|---|
| | Local | Global | Local | Global |
| FedAvg | 25.58 | 26.10 | 13.48 | 12.87 |
| IFCA | 31.90 | 10.47 | 18.34 | 12.00 |
| CFL | 26.06 | 24.53 | 13.94 | 12.60 |
| CFL (3) | 25.18 | 24.80 | 13.34 | 12.13 |
| FeSEM | 34.56 | 21.93 | 17.28 | 12.90 |
| FedEM | 41.28 | 34.17 | 25.82 | 24.77 |
| FedRC | **49.16** | **47.80** | **31.76** | **28.10** |

concepts increases; 2) When clients have only no concept shifts, employing multiple clusters (such as FedEM and FeSEM) does not perform as well as using a single model (FedAvg). Nonetheless, FedRC outperforms FedAvg and retains robustness.

**Effectiveness of FedRC remains when using hard clustering.** The prediction of FedRC requires the ensemble of $K$ clusters (soft clustering). As most existing clustered FL methods are hard clustering methods (Long et al., 2023; Ghosh et al., 2020; Sattler et al., 2020b), we also evaluate the performance of FedRC when using hard clustering here, i.e., only using the cluster with the largest clustering weights $\omega_{i;k}$ for prediction. Figure 6(c) shows that using hard clustering with not affect the effectiveness of FedRC.

## 6 CONCLUSION, LIMITATIONS, AND FUTURE WORKS

This paper addresses the diverse distribution shift challenge in FL and proposes using clustered FL methods to tackle it. However, we found that none of the existing clustered FL methods effectively address the diverse distribution shift challenge, leading us to introduce FedRC as a solution. Furthermore, we have explored extensions in Appendix I and Appendix F, including improvements in communication and computation efficiency, automatic determination of the number of clusters, and mitigation of the personalization-generalization trade-offs. For future research, delving deeper into providing a more comprehensive theoretical understanding of the distinctions in clustering results between FedRC and other methods would be intriguing.

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

Contents of Appendix

A    Proof of Optimization Steps

**Theorem A.1.** *When maximizing Equation 2, the optimization steps are given by,*
***Optimizing*** $\Omega$*:*

$$\gamma_{i,j;k}^t = \frac{\omega_{i;k}^{t-1}\tilde{\mathcal{I}}(\mathbf{x}_{ij}, y_{ij}, \boldsymbol{\theta}_k)}{\sum_{i=1}^K \omega_i^{t-1}\tilde{\mathcal{I}}(\mathbf{x}_{ij}, y_{ij}, \boldsymbol{\theta}_k)} \,, \tag{7}$$

$$\omega_{i;k}^t = \frac{1}{N_i}\sum_{j=1}^{N_i} \gamma_{i,j;k}^t \,. \tag{8}$$

***Optimizing*** $\Theta$*:*

$$\tag{9}$$

$$\boldsymbol{\theta}_k^t = \boldsymbol{\theta}_k^{t-1} - \frac{\eta}{N}\sum_{i=1}^M\sum_{j=1}^{N_i} \gamma_{i,j;k}^t \nabla_{\boldsymbol{\theta}_k^{t-1}} f(\mathbf{x}_{ij}, y_{ij}, \boldsymbol{\theta}_k^{t-1}) \,, \tag{10}$$

*Proof.* Consider the objective function,

$$\mathcal{L}(\boldsymbol{\Theta}, \boldsymbol{\Omega}) = \frac{1}{N}\sum_{i=1}^M\sum_{j=1}^N \ln\left(\sum_{k=1}^K \omega_{i;k}\tilde{\mathcal{I}}(\mathbf{x}_{ij}, y_{ij}; \boldsymbol{\theta}_k)\right) + \sum_{i=1}^M \lambda_i\left(\sum_{k=1}^K \omega_{i;k} - 1\right), \tag{11}$$

Taking the derivative of $\mathcal{L}(\boldsymbol{\Theta})$, we have,

$$\frac{\partial \mathcal{L}(\boldsymbol{\Theta}, \boldsymbol{\Omega})}{\partial \omega_{i;k}} = \frac{1}{N} \sum_{j=1}^{N_i} \frac{\tilde{\mathcal{I}}(\mathbf{x}_{ij}, y_{ij}; \boldsymbol{\theta}_k)}{\sum_{n=1}^{K} \omega_{i;n} \tilde{\mathcal{I}}_n(\mathbf{x}_{ij}, y_{ij}; \boldsymbol{\theta}_n)} + \lambda_i, \tag{12}$$

define

$$\gamma_{i,j;k} = \frac{\omega_{i;k} \tilde{\mathcal{I}}(\mathbf{x}_{ij}, y_{ij}; \boldsymbol{\theta}_k)}{\sum_{n=1}^{K} \omega_{i;n} \tilde{\mathcal{I}}_n(\mathbf{x}_{ij}, y_{ij}; \boldsymbol{\theta}_n)}, \tag{13}$$

and set $\frac{\partial \mathcal{L}(\boldsymbol{\Theta})}{\partial \omega_{i;k}} = 0$ to obtain the optimal $\omega_{i;k}$, we have,

$$\frac{1}{N} \sum_{j=1}^{N_i} \frac{\gamma_{i,j;k}}{\omega_{i;k}} = -\lambda_i, \tag{14}$$

$$\omega_{i;k} = \frac{1}{-N\lambda_i} \sum_{j=1}^{N_i} \gamma_{i,j;k}. \tag{15}$$

By setting $\sum_{k=1}^{K} \omega_{i;k} = 1$ and considering that $\sum_{k=1}^{K} \gamma_{i,j;k} = 1$, we directly derive the result $\lambda_i = \frac{-N_i}{N}$. Then we have

$$\omega_{i;k} = \frac{1}{N_i} \sum_{j=1}^{N_i} \gamma_{i,j;k}. \tag{16}$$

Then consider to optimize $\boldsymbol{\theta}_k$, we have,

$$\frac{\partial \mathcal{L}(\boldsymbol{\Theta}, \boldsymbol{\Omega})}{\partial \boldsymbol{\theta}_k} = \frac{1}{N} \sum_{i=1}^{M} \sum_{j=1}^{N_i} \frac{\omega_{i;k}}{\sum_{n=1}^{K} \omega_{i;n} \tilde{\mathcal{I}}_n(\mathbf{x}_{ij}, y_{ij}; \boldsymbol{\theta}_n)} \cdot \frac{\partial \tilde{\mathcal{I}}(\mathbf{x}_{ij}, y_{ij}; \boldsymbol{\theta}_k)}{\partial \boldsymbol{\theta}_k}, \tag{17}$$

$$= \frac{1}{N} \sum_{i=1}^{M} \sum_{j=1}^{N_i} \frac{\omega_{i;k}}{\sum_{n=1}^{K} \omega_{i;n} \tilde{\mathcal{I}}_n(\mathbf{x}_{ij}, y_{ij}; \boldsymbol{\theta}_n)} \cdot \frac{\exp(-f(\mathbf{x}_{ij}, y_{ij}, \boldsymbol{\theta}_k)) \sum_{i=1}^{M} \sum_{j=1}^{N} \gamma_{i,j;k}}{\sum_{i=1}^{M} \sum_{j=1}^{N} \mathbf{1}_{y_{ij}=y} \gamma_{i,j;k}}$$
$$\cdot (-\nabla_{\boldsymbol{\theta}_k} f(\mathbf{x}_{ij}, y_{ij}, \boldsymbol{\theta}_k)), \tag{18}$$

$$= -\frac{1}{N} \sum_{i=1}^{M} \sum_{j=1}^{N_i} \gamma_{i,j;k} \nabla_{\boldsymbol{\theta}_k} f(\mathbf{x}_{ij}, y_{ij}, \boldsymbol{\theta}_k). \tag{19}$$

Because if hard to find a close-form solution to $\frac{\partial \mathcal{L}(\boldsymbol{\Theta}, \boldsymbol{\Omega})}{\partial \boldsymbol{\theta}_k} = 0$ when $\boldsymbol{\theta}_k$ is the parameter of deep neural networks, we use gradient ascent to optimize $\boldsymbol{\theta}_k$. Then we finish the proof of optimization steps. $\quad \square$

## B  PROOF OF THEOREM 4.3

**Lemma B.1.** *Define*

$$h(\boldsymbol{\theta}_x) = \frac{\omega_{i;k} \tilde{\mathcal{I}}(\mathbf{x}_{ij}, y_{ij}, \boldsymbol{\theta}_x)}{\sum_{n=1}^{K} \omega_{i;n} \tilde{\mathcal{I}}(\mathbf{x}_{ij}, y_{ij}, \boldsymbol{\theta}_n)}, \tag{20}$$

*which corresponding to* $\boldsymbol{\Theta} = [\boldsymbol{\theta}_1, \cdots, \boldsymbol{\theta}_x, \boldsymbol{\theta}_{k+1}, \cdots, \boldsymbol{\theta}_K]$, *we have,*

$$|h(\boldsymbol{\theta}_x) - h(\boldsymbol{\theta}_y)| \leq \frac{L}{8} \|\boldsymbol{\theta}_1 - \boldsymbol{\theta}_2\|^2 + \frac{3}{8} \|\nabla f(\mathbf{x}_{ij}, y_{ij}, \boldsymbol{\theta}_y)\| \|\boldsymbol{\theta}_1 - \boldsymbol{\theta}_2\|. \tag{21}$$

*Proof.* Define,

$$h(\boldsymbol{\theta}_x) = \frac{\omega_{i;k} \tilde{\mathcal{I}}(\mathbf{x}_{ij}, y_{ij}, \boldsymbol{\theta}_x)}{\sum_{n=1}^{K} \omega_{i;n} \tilde{\mathcal{I}}(\mathbf{x}_{ij}, y_{ij}, \boldsymbol{\theta}_n)}, \tag{22}$$

$$= \frac{\tilde{\omega}_{i;k} \exp(-f(\mathbf{x}_{ij}, y_{ij}, \boldsymbol{\theta}_x))}{\sum_{n=1}^{K} \tilde{\omega}_{i;n} \exp(-f(\mathbf{x}_{ij}, y_{ij}, \boldsymbol{\theta}_n))}, \tag{23}$$

where $\tilde{\omega}_{i;k} = \frac{\omega_{i;k}}{\mathcal{P}(y=y_{ij};\boldsymbol{\theta}_k)}$. Then we have,

$$
\begin{aligned}
\frac{\partial h(\boldsymbol{\theta}_x)}{\partial \boldsymbol{\theta}_x} &= \frac{-\tilde{\omega}_{i;k}\exp(-f(\mathbf{x}_{ij},y_{ij},\boldsymbol{\theta}_x))\nabla f(\mathbf{x}_{ij},y_{ij},\boldsymbol{\theta}_x)}{\sum_{n=1}^{K}\tilde{\omega}_{i;n}\exp(-f(\mathbf{x}_{ij},y_{ij},\boldsymbol{\theta}_n))} \\
&\quad - \frac{-\tilde{\omega}_{i;k}^2\exp^2(-f(\mathbf{x}_{ij},y_{ij},\boldsymbol{\theta}_x))\nabla f(\mathbf{x}_{ij},y_{ij},\boldsymbol{\theta}_x)}{\left(\sum_{n=1}^{K}\tilde{\omega}_{i;n}\exp(-f(\mathbf{x}_{ij},y_{ij},\boldsymbol{\theta}_n))\right)^2}, 
\end{aligned}
\tag{24}
$$

$$
= \nabla f(\mathbf{x}_{ij},y_{ij},\boldsymbol{\theta}_x)h(\boldsymbol{\theta}_x)\left(-1+h(\boldsymbol{\theta}_x)\right). \tag{25}
$$

Then we have,

$$
\|\nabla h(\boldsymbol{\theta}_x)-\nabla h(\boldsymbol{\theta}_y)\| = \|h(\boldsymbol{\theta}_x)\left(1-h(\boldsymbol{\theta}_x)\right)\nabla f(\mathbf{x}_{ij},y_{ij},\boldsymbol{\theta}_x)-h(\boldsymbol{\theta}_y)\left(1-h(\boldsymbol{\theta}_y)\right)\nabla f(\mathbf{x}_{ij},y_{ij},\boldsymbol{\theta}_y)\|, \tag{26}
$$

$$
\leq \frac{L}{4}\|\boldsymbol{\theta}_x-\boldsymbol{\theta}_y\| + \frac{1}{4}\|\nabla f(\mathbf{x}_{ij},y_{ij},\boldsymbol{\theta}_y)\|. \tag{27}
$$

On the other hand, for $h(\boldsymbol{\theta}_x)$, we can always find that,

$$
h(\boldsymbol{\theta}_x) = h(\boldsymbol{\theta}_y) + \int_0^1 \langle \nabla h(\boldsymbol{\theta}_y+\tau(\boldsymbol{\theta}_x-\boldsymbol{\theta}_y)), \boldsymbol{\theta}_x-\boldsymbol{\theta}_y\rangle\, d\tau, \tag{28}
$$

$$
= h(\boldsymbol{\theta}_y) + \langle \nabla h(\boldsymbol{\theta}_y),\boldsymbol{\theta}_x-\boldsymbol{\theta}_y\rangle + \int_0^1 \langle \nabla h(\boldsymbol{\theta}_y+\tau(\boldsymbol{\theta}_x-\boldsymbol{\theta}_y))-\nabla h(\boldsymbol{\theta}_y),\boldsymbol{\theta}_x-\boldsymbol{\theta}_y\rangle\, d\tau. \tag{29}
$$

Then we have,

$$
\begin{aligned}
&|h(\boldsymbol{\theta}_x)-h(\boldsymbol{\theta}_y)-\langle\nabla h(\boldsymbol{\theta}_y),\boldsymbol{\theta}_x-\boldsymbol{\theta}_y\rangle| \\
&= \left|\int_0^1 \langle\nabla h(\boldsymbol{\theta}_y+\tau(\boldsymbol{\theta}_x-\boldsymbol{\theta}_y))-\nabla h(\boldsymbol{\theta}_y),\boldsymbol{\theta}_x-\boldsymbol{\theta}_y\rangle\, d\tau\right|,
\end{aligned}
\tag{30}
$$

$$
\leq \int_0^1 |\langle\nabla h(\boldsymbol{\theta}_y+\tau(\boldsymbol{\theta}_x-\boldsymbol{\theta}_y))-\nabla h(\boldsymbol{\theta}_y),\boldsymbol{\theta}_x-\boldsymbol{\theta}_y\rangle|\, d\tau, \tag{31}
$$

$$
\leq \int_0^1 \|\nabla h(\boldsymbol{\theta}_y+\tau(\boldsymbol{\theta}_x-\boldsymbol{\theta}_y))-\nabla h(\boldsymbol{\theta}_y)\|\,\|\boldsymbol{\theta}_x-\boldsymbol{\theta}_y\|\, d\tau, \tag{32}
$$

$$
\leq \int_0^1 \tau\left(\frac{L}{4}\|\boldsymbol{\theta}_x-\boldsymbol{\theta}_y\|^2 + \frac{1}{4}\|\nabla f(\mathbf{x}_{ij},y_{ij},\boldsymbol{\theta}_y)\|\,\|\boldsymbol{\theta}_1-\boldsymbol{\theta}_2\|\right)\, d\tau, \tag{33}
$$

$$
= \frac{L}{8}\|\boldsymbol{\theta}_x-\boldsymbol{\theta}_y\|^2 + \frac{1}{8}\|\nabla f(\mathbf{x}_{ij},y_{ij},\boldsymbol{\theta}_y)\|\,\|\boldsymbol{\theta}_x-\boldsymbol{\theta}_y\|. \tag{34}
$$

Therefore, we have,

$$
|h(\boldsymbol{\theta}_x)-h(\boldsymbol{\theta}_y)| \leq \frac{L}{8}\|\boldsymbol{\theta}_1-\boldsymbol{\theta}_2\|^2 + \frac{3}{8}\|\nabla f(\mathbf{x}_{ij},y_{ij},\boldsymbol{\theta}_y)\|\,\|\boldsymbol{\theta}_x-\boldsymbol{\theta}_y\|. \tag{35}
$$

$\square$

**Lemma B.2.** *Assume $f(x,y,\boldsymbol{\theta})$ is $L$-smooth (Assumption 1), define,*

$$
g(\boldsymbol{\theta}_k) = \frac{1}{N}\sum_{i=1}^{M}\sum_{j=1}^{N_i}\ln\left(\sum_{n=1}^{K}\omega_{i;n}\tilde{\mathcal{I}}_n(\mathbf{x}_{ij},y_{ij};\boldsymbol{\theta}_n)\right), \tag{36}
$$

*where $1 \leq k \leq K$. Then we have,*

$$
\|\nabla g(\boldsymbol{\theta}_1)-\nabla g(\boldsymbol{\theta}_2)\|
$$

$$
\leq L\|\boldsymbol{\theta}_1-\boldsymbol{\theta}_2\| + \frac{1}{N}\sum_{i=1}^{M}\sum_{j=1}^{N_i}\left(\frac{L}{8}\|\boldsymbol{\theta}_1-\boldsymbol{\theta}_2\|^2\|\nabla f(\mathbf{x}_{ij},y_{ij},\boldsymbol{\theta}_2)\| + \frac{3}{8}\|\boldsymbol{\theta}_1-\boldsymbol{\theta}_2\|\|\nabla f(\mathbf{x}_{ij},y_{ij},\boldsymbol{\theta}_2)\|^2\right),
$$

$$
\tag{37}
$$

where $\gamma_{i,j;k}^1$ and $\gamma_{i,j;k}^2$ are defined in Theorem A.1 corresponding to $\boldsymbol{\Theta}_1$ and $\boldsymbol{\Theta}_2$ respectively. We can further prove that,

$$
\begin{aligned}
g(\boldsymbol{\theta}_1) \leq{} & g(\boldsymbol{\theta}_2) + \langle \nabla g(\boldsymbol{\theta}_2), \boldsymbol{\theta}_1 - \boldsymbol{\theta}_2 \rangle \\
& + \left( \frac{L}{2} + \frac{3}{16N} \sum_{i=1}^{M} \sum_{j=1}^{N_i} \|\nabla f(\mathbf{x}_{ij}, y_{ij}, \boldsymbol{\theta}_2)\|^2 \right) \|\boldsymbol{\theta}_1 - \boldsymbol{\theta}_2\|^2 \\
& + \left( \frac{L}{16N} \sum_{i=1}^{M} \sum_{j=1}^{N_i} \|\nabla f(\mathbf{x}_{ij}, y_{ij}, \boldsymbol{\theta}_2)\| \right) \|\boldsymbol{\theta}_1 - \boldsymbol{\theta}_2\|^3 \,,
\end{aligned}
\tag{38}
$$

$$
\begin{aligned}
g(\boldsymbol{\theta}_1) \geq{} & g(\boldsymbol{\theta}_2) + \langle \nabla g(\boldsymbol{\theta}_2), \boldsymbol{\theta}_1 - \boldsymbol{\theta}_2 \rangle \\
& - \left( \frac{L}{2} + \frac{3}{16N} \sum_{i=1}^{M} \sum_{j=1}^{N_i} \|\nabla f(\mathbf{x}_{ij}, y_{ij}, \boldsymbol{\theta}_2)\|^2 \right) \|\boldsymbol{\theta}_1 - \boldsymbol{\theta}_2\|^2 \\
& - \left( \frac{L}{16N} \sum_{i=1}^{M} \sum_{j=1}^{N_i} \|\nabla f(\mathbf{x}_{ij}, y_{ij}, \boldsymbol{\theta}_2)\| \right) \|\boldsymbol{\theta}_1 - \boldsymbol{\theta}_2\|^3 \,.
\end{aligned}
\tag{39}
$$

*Proof.* Based on the results in Theorem A.1, Section A, we have,

$$
\frac{\partial g(\boldsymbol{\theta}_k)}{\partial \boldsymbol{\theta}_k} = -\frac{1}{N} \sum_{i=1}^{M} \sum_{j=1}^{N_i} \gamma_{i,j;k} \nabla_{\boldsymbol{\theta}_k} f(\mathbf{x}_{ij}, y_{ij}, \boldsymbol{\theta}_k) \,,
\tag{40}
$$

then we have,

$$
\|\nabla g(\boldsymbol{\theta}_1) - \nabla g(\boldsymbol{\theta}_2)\|
$$

$$
= \left\| \frac{1}{N} \sum_{i=1}^{M} \sum_{j=1}^{N_i} \gamma_{i,j;k}^1 \nabla f(\mathbf{x}_{ij}, y_{ij}, \boldsymbol{\theta}_2) - \frac{1}{N} \sum_{i=1}^{M} \sum_{j=1}^{N_i} \gamma_{i,j;k}^2 \nabla f(\mathbf{x}_{ij}, y_{ij}, \boldsymbol{\theta}_1) \right\| \,,
\tag{41}
$$

$$
\leq \frac{1}{N} \sum_{i=1}^{M} \sum_{j=1}^{N_i} \left\| \gamma_{i,j;k}^1 \nabla f(\mathbf{x}_{ij}, y_{ij}, \boldsymbol{\theta}_2) - \gamma_{i,j;k}^2 \nabla f(\mathbf{x}_{ij}, y_{ij}, \boldsymbol{\theta}_1) \right\| \,,
\tag{42}
$$

$$
\leq \frac{1}{N} \sum_{i=1}^{M} \sum_{j=1}^{N_i} \gamma_{i,j;k}^1 \left\| \nabla f(\mathbf{x}_{ij}, y_{ij}, \boldsymbol{\theta}_2) - \nabla f(\mathbf{x}_{ij}, y_{ij}, \boldsymbol{\theta}_1) \right\|
$$

$$
+ \frac{1}{N} \sum_{i=1}^{M} \sum_{j=1}^{N_i} |\gamma_{i,j;k}^1 - \gamma_{i,j;k}^2| \left\| \nabla f(\mathbf{x}_{ij}, y_{ij}, \boldsymbol{\theta}_2) \right\| \,,
\tag{43}
$$

$$
\leq L \|\boldsymbol{\theta}_1 - \boldsymbol{\theta}_2\|
$$

$$
+ \frac{1}{N} \sum_{i=1}^{M} \sum_{j=1}^{N_i} \left( \frac{L}{8} \|\boldsymbol{\theta}_1 - \boldsymbol{\theta}_2\|^2 \|\nabla f(\mathbf{x}_{ij}, y_{ij}, \boldsymbol{\theta}_2)\| + \frac{3}{8} \|\boldsymbol{\theta}_1 - \boldsymbol{\theta}_2\| \|\nabla f(\mathbf{x}_{ij}, y_{ij}, \boldsymbol{\theta}_2)\|^2 \right).
\tag{44}
$$

On the other hand, for $f(\boldsymbol{\theta}_k)$, we can always find that,

$$
g(\boldsymbol{\theta}_1) = g(\boldsymbol{\theta}_2) + \int_0^1 \langle \nabla g(\boldsymbol{\theta}_2 + \tau(\boldsymbol{\theta}_1 - \boldsymbol{\theta}_2)), \boldsymbol{\theta}_1 - \boldsymbol{\theta}_2 \rangle \, d\tau \,,
\tag{45}
$$

$$
= g(\boldsymbol{\theta}_2) + \langle \nabla g(\boldsymbol{\theta}_2), \boldsymbol{\theta}_1 - \boldsymbol{\theta}_2 \rangle + \int_0^1 \langle \nabla g(\boldsymbol{\theta}_2 + \tau(\boldsymbol{\theta}_1 - \boldsymbol{\theta}_2)) - \nabla g(\boldsymbol{\theta}_2), \boldsymbol{\theta}_1 - \boldsymbol{\theta}_2 \rangle \, d\tau.
\tag{46}
$$

Then we have,

$$|g(\boldsymbol{\theta}_1) - g(\boldsymbol{\theta}_2) - \langle \nabla g(\boldsymbol{\theta}_2), \boldsymbol{\theta}_1 - \boldsymbol{\theta}_2 \rangle|$$

$$= \left| \int_0^1 \langle \nabla g(\boldsymbol{\theta}_2 + \tau(\boldsymbol{\theta}_1 - \boldsymbol{\theta}_2)) - \nabla g(\boldsymbol{\theta}_2), \boldsymbol{\theta}_1 - \boldsymbol{\theta}_2 \rangle \, d\tau \right| , \tag{47}$$

$$\leq \int_0^1 |\langle \nabla g(\boldsymbol{\theta}_2 + \tau(\boldsymbol{\theta}_1 - \boldsymbol{\theta}_2)) - \nabla g(\boldsymbol{\theta}_2), \boldsymbol{\theta}_1 - \boldsymbol{\theta}_2 \rangle| \, d\tau , \tag{48}$$

$$\leq \int_0^1 \|\nabla g(\boldsymbol{\theta}_2 + \tau(\boldsymbol{\theta}_1 - \boldsymbol{\theta}_2)) - \nabla g(\boldsymbol{\theta}_2)\| \, \|\boldsymbol{\theta}_1 - \boldsymbol{\theta}_2\| \, d\tau , \tag{49}$$

$$\leq \int_0^1 \tau \left( \left( L + \frac{3}{8N} \sum_{i=1}^M \sum_{j=1}^{N_i} \|\nabla f(\mathbf{x}_{ij}, y_{ij}, \boldsymbol{\theta}_2)\|^2 \right) \|\boldsymbol{\theta}_1 - \boldsymbol{\theta}_2\|^2 \right) d\tau$$

$$+ \int_0^1 \tau \left( \left( \frac{L}{8N} \sum_{i=1}^M \sum_{j=1}^{N_i} \|\nabla f(\mathbf{x}_{ij}, y_{ij}, \boldsymbol{\theta}_2)\| \right) \|\boldsymbol{\theta}_1 - \boldsymbol{\theta}_2\|^3 \right) d\tau , \tag{50}$$

$$= \left( \frac{L}{2} + \frac{3}{16N} \sum_{i=1}^M \sum_{j=1}^{N_i} \|\nabla f(\mathbf{x}_{ij}, y_{ij}, \boldsymbol{\theta}_2)\|^2 \right) \|\boldsymbol{\theta}_1 - \boldsymbol{\theta}_2\|^2$$

$$+ \left( \frac{L}{16N} \sum_{i=1}^M \sum_{j=1}^{N_i} \|\nabla f(\mathbf{x}_{ij}, y_{ij}, \boldsymbol{\theta}_2)\| \right) \|\boldsymbol{\theta}_1 - \boldsymbol{\theta}_2\|^3 . \tag{51}$$

Then we have,

$$g(\boldsymbol{\theta}_1) \leq g(\boldsymbol{\theta}_2) + \langle \nabla g(\boldsymbol{\theta}_2), \boldsymbol{\theta}_1 - \boldsymbol{\theta}_2 \rangle$$

$$+ \left( \frac{L}{2} + \frac{3}{16N} \sum_{i=1}^M \sum_{j=1}^{N_i} \|\nabla f(\mathbf{x}_{ij}, y_{ij}, \boldsymbol{\theta}_2)\|^2 \right) \|\boldsymbol{\theta}_1 - \boldsymbol{\theta}_2\|^2$$

$$+ \left( \frac{L}{16N} \sum_{i=1}^M \sum_{j=1}^{N_i} \|\nabla f(\mathbf{x}_{ij}, y_{ij}, \boldsymbol{\theta}_2)\| \right) \|\boldsymbol{\theta}_1 - \boldsymbol{\theta}_2\|^3 , \tag{52}$$

$$g(\boldsymbol{\theta}_1) \geq g(\boldsymbol{\theta}_2) + \langle \nabla g(\boldsymbol{\theta}_2), \boldsymbol{\theta}_1 - \boldsymbol{\theta}_2 \rangle$$

$$- \left( \frac{L}{2} + \frac{3}{16N} \sum_{i=1}^M \sum_{j=1}^{N_i} \|\nabla f(\mathbf{x}_{ij}, y_{ij}, \boldsymbol{\theta}_2)\|^2 \right) \|\boldsymbol{\theta}_1 - \boldsymbol{\theta}_2\|^2$$

$$- \left( \frac{L}{16N} \sum_{i=1}^M \sum_{j=1}^{N_i} \|\nabla f(\mathbf{x}_{ij}, y_{ij}, \boldsymbol{\theta}_2)\| \right) \|\boldsymbol{\theta}_1 - \boldsymbol{\theta}_2\|^3 . \tag{53}$$

$\square$

**Theorem B.3** (Convergence rate of RobustCluster). *Assume $f$ is L-smooth (Assumption 1), setting $T$ as the number of iterations, and $\eta = \frac{8}{40L + 9\sigma^2}$, we have,*

$$\frac{1}{T} \sum_{t=0}^{T-1} \sum_{k=1}^K \left\| \nabla_{\boldsymbol{\theta}_k} \mathcal{L}\left(\boldsymbol{\Theta}^t\right) \right\|^2 \leq O\left( \frac{(40L + 9\sigma^2)\left(\mathcal{L}(\boldsymbol{\Theta}^*) - \mathcal{L}(\boldsymbol{\Theta}^0)\right)}{4T} \right) , \tag{54}$$

*which denotes the algorithm achieve sub-linear convergence rate.*

*Proof.* Each M step of RobustCluster can be seen as optimizing $\boldsymbol{\theta}_1, \cdots, \boldsymbol{\theta}_K$ respectively. Then we have,

$$\mathcal{L}(\boldsymbol{\Theta}^{t+1}) - \mathcal{L}(\boldsymbol{\Theta}^t) = \sum_{k=1}^K \mathcal{L}(\boldsymbol{\Theta}_k^t) - \mathcal{L}(\boldsymbol{\Theta}_{k-1}^t) , \tag{55}$$

where we define,

$$\boldsymbol{\Theta}_k^t = \left[ \boldsymbol{\theta}_1^{t+1}, \cdots, \boldsymbol{\theta}_k^{t+1}, \boldsymbol{\theta}_{k+1}^t \cdots, \boldsymbol{\theta}_K^t \right] , \tag{56}$$

and $\boldsymbol{\Theta}_0^t = \boldsymbol{\Theta}^t$, $\boldsymbol{\Theta}_K^t = \boldsymbol{\Theta}^{t+1}$. Then we define,

$$F_k(\boldsymbol{\theta}_\nu) = \frac{1}{N} \sum_{i=1}^{M} \sum_{j=1}^{N_i} \ln \left( \sum_{n=1}^{K} \omega_{i;n} \tilde{\mathcal{I}}_{in}(\mathbf{x}_{ij}, y_{ij}; \boldsymbol{\theta}_n) \right), \tag{57}$$

where $\boldsymbol{\theta}_n \in \boldsymbol{\Theta}_k^t$ for $n \neq \nu$, and $\boldsymbol{\theta}_\nu$ is the variable in $F_k(\boldsymbol{\theta}_\nu)$. Define $\gamma_{i,j;k}^k$ that corresponding to $\boldsymbol{\Theta}_k$, we have,

$$\mathcal{L}(\boldsymbol{\Theta}^{t+1}) - \mathcal{L}(\boldsymbol{\Theta}^t)$$

$$= \sum_{k=1}^{K} \mathcal{L}(\boldsymbol{\Theta}_k^t) - \mathcal{L}(\boldsymbol{\Theta}_{k-1}^t), \tag{58}$$

$$= \sum_{k=1}^{K} F_k(\boldsymbol{\theta}_k^{t+1}) - F_k(\boldsymbol{\theta}_k^t), \tag{59}$$

$$\geq \sum_{k=1}^{K} \left\langle \nabla F_k(\boldsymbol{\theta}_k^t), \boldsymbol{\theta}_k^{t+1} - \boldsymbol{\theta}_k^t \right\rangle,$$

$$- \left( \frac{L}{2} + \frac{3}{16N} \sum_{i=1}^{M} \sum_{j=1}^{N_i} \left\| \nabla f(\mathbf{x}_{ij}, y_{ij}, \boldsymbol{\theta}_k^t) \right\|^2 \right) \left\| \boldsymbol{\theta}_k^{t+1} - \boldsymbol{\theta}_k^t \right\|^2$$

$$- \left( \frac{L}{16N} \sum_{i=1}^{M} \sum_{j=1}^{N_i} \left\| \nabla f(\mathbf{x}_{ij}, y_{ij}, \boldsymbol{\theta}_k^t) \right\| \right) \left\| \boldsymbol{\theta}_k^{t+1} - \boldsymbol{\theta}_k^t \right\|^3. \tag{60}$$

The last inequality cones from Lemma B.2. From the results in Theorem A.1, Section A, we have,

$$\nabla F_k(\boldsymbol{\theta}_k^t) = -\frac{1}{N} \sum_{i=1}^{M} \sum_{j=1}^{N_i} \gamma_{i,j;k}^k \nabla_{\boldsymbol{\theta}_k} f(\mathbf{x}_{ij}, y_{ij}, \boldsymbol{\theta}_k), \tag{61}$$

at the same time, we have,

$$\boldsymbol{\theta}_k^{t+1} - \boldsymbol{\theta}_k^t = \eta \nabla F_0(\boldsymbol{\theta}_k^t) = -\frac{\eta}{N} \sum_{i=1}^{M} \sum_{j=1}^{N_i} \gamma_{i,j;k}^0 \nabla_{\boldsymbol{\theta}_k} f(\mathbf{x}_{ij}, y_{ij}, \boldsymbol{\theta}_k). \tag{62}$$

Then we can obtain,

$$\left\langle \nabla F_k(\boldsymbol{\theta}_k^t), \boldsymbol{\theta}_k^{t+1} - \boldsymbol{\theta}_k^t \right\rangle$$

$$= \left\langle \nabla F_k(\boldsymbol{\theta}_k^t) - \nabla F_0(\boldsymbol{\theta}_k^t) + \nabla F_0(\boldsymbol{\theta}_k^t), \boldsymbol{\theta}_k^{t+1} - \boldsymbol{\theta}_k^t \right\rangle, \tag{63}$$

$$= \left\langle \nabla F_k(\boldsymbol{\theta}_k^t) - \nabla F_0(\boldsymbol{\theta}_k^t), \boldsymbol{\theta}_k^{t+1} - \boldsymbol{\theta}_k^t \right\rangle + \left\langle \nabla F_0(\boldsymbol{\theta}_k^t), \boldsymbol{\theta}_k^{t+1} - \boldsymbol{\theta}_k^t \right\rangle, \tag{64}$$

$$\geq - \left\| \nabla F_k(\boldsymbol{\theta}_k^t) - \nabla F_0(\boldsymbol{\theta}_k^t) \right\| \left\| \boldsymbol{\theta}_k^{t+1} - \boldsymbol{\theta}_k^t \right\| + \frac{1}{\eta} \left\| \boldsymbol{\theta}_k^{t+1} - \boldsymbol{\theta}_k^t \right\|^2, \tag{65}$$

$$\geq \left( \frac{1}{\eta} - 2L - \frac{3}{8N} \sum_{i=1}^{M} \sum_{j=1}^{N_i} \left\| \nabla f(\mathbf{x}_{ij}, y_{ij}, \boldsymbol{\theta}_k^t) \right\|^2 \right) \left\| \boldsymbol{\theta}_k^{t+1} - \boldsymbol{\theta}_k^t \right\|^2$$

$$- \left( \frac{L}{8N} \sum_{i=1}^{M} \sum_{j=1}^{N_i} \left\| \nabla f(\mathbf{x}_{ij}, y_{ij}, \boldsymbol{\theta}_k^t) \right\| \right) \left\| \boldsymbol{\theta}_k^{t+1} - \boldsymbol{\theta}_k^t \right\|^3. \tag{66}$$

Combine Equation (60) and Equation (66), we have,

$$
\mathcal{L}(\boldsymbol{\Theta}^{t+1}) - \mathcal{L}(\boldsymbol{\Theta}^{t})
$$

$$
\geq \sum_{k=1}^{K} \left( \frac{1}{\eta} - \frac{5L}{2} - \frac{9}{16N} \sum_{i=1}^{M} \sum_{j=1}^{N_i} \left\| \nabla f(\mathbf{x}_{ij}, y_{ij}, \boldsymbol{\theta}_k^t) \right\|^2 \right) \left\| \boldsymbol{\theta}_k^{t+1} - \boldsymbol{\theta}_k^t \right\|^2
$$

$$
- \left( \frac{3L}{16N} \sum_{i=1}^{M} \sum_{j=1}^{N_i} \left\| \nabla f(\mathbf{x}_{ij}, y_{ij}, \boldsymbol{\theta}_k^t) \right\| \right) \left\| \boldsymbol{\theta}_k^{t+1} - \boldsymbol{\theta}_k^t \right\|^3 , \tag{67}
$$

$$
\geq \sum_{k=1}^{K} \left( \frac{1}{\eta} - \frac{5L}{2} - \frac{9\sigma^2}{16} \right) \left\| \boldsymbol{\theta}_k^{t+1} - \boldsymbol{\theta}_k^t \right\|^2
$$

$$
- \frac{3L}{16N} \left( \sum_{i=1}^{M} \sum_{j=1}^{N_i} \left\| \nabla f(\mathbf{x}_{ij}, y_{ij}, \boldsymbol{\theta}_k^t) \right\| \right) \left\| \frac{\eta}{N} \sum_{i=1}^{M} \sum_{j=1}^{N_i} \gamma_{i,j;k}^0 \nabla f(\mathbf{x}_{ij}, y_{ij}, \boldsymbol{\theta}_k^t) \right\| \left\| \boldsymbol{\theta}_k^{t+1} - \boldsymbol{\theta}_k^t \right\|^2 ,
$$

$$
\tag{68}
$$

$$
\geq \sum_{k=1}^{K} \left( \frac{1}{\eta} - \frac{5L}{2} - \frac{3\sigma^2}{16} \right) \left\| \boldsymbol{\theta}_k^{t+1} - \boldsymbol{\theta}_k^t \right\|^2 - \frac{3\eta L}{16} \left( \frac{1}{N} \sum_{i=1}^{M} \sum_{j=1}^{N_i} \left\| \nabla f(\mathbf{x}_{ij}, y_{ij}, \boldsymbol{\theta}_k^t) \right\| \right)^2 \left\| \boldsymbol{\theta}_k^{t+1} - \boldsymbol{\theta}_k^t \right\|^2 ,
$$

$$
\tag{69}
$$

$$
\geq \sum_{k=1}^{K} \left( \frac{1}{\eta} - \frac{5L}{2} - \frac{9\sigma^2}{16} - \frac{3\eta L\sigma^2}{16} \right) \left\| \boldsymbol{\theta}_k^{t+1} - \boldsymbol{\theta}_k^t \right\|^2 , \tag{70}
$$

$$
= \sum_{k=1}^{K} \left( \eta - \frac{5L\eta^2}{2} - \frac{9\sigma^2\eta^2}{16} - \frac{3\eta^3 L\sigma^2}{16} \right) \left\| \nabla_{\boldsymbol{\theta}_k} \mathcal{L}(\boldsymbol{\Theta}^t) \right\|^2 . \tag{71}
$$

Then we can observe that the objective $\mathcal{L}(\boldsymbol{\Theta})$ converge when,

$$
\eta \leq \frac{((1600L^2 + 912L\sigma^2 + 81\sigma^4)^{1/2} - 40L - 9\sigma^2}{6L\sigma^2} , \tag{72}
$$

and when we set,

$$
\eta \leq \frac{((1600L^2 + 816L\sigma^2 + 81\sigma^4)^{1/2} - 40L - 9\sigma^2}{6L\sigma^2} , \tag{73}
$$

$$
= \frac{\sqrt{(40L + 9\sigma^2)^2 + 96L\sigma^2} - (40L + 9\sigma^2)}{6L\sigma^2} , \tag{74}
$$

we have,

$$
\mathcal{L}(\boldsymbol{\Theta}^{t+1}) - \mathcal{L}(\boldsymbol{\Theta}^{t}) \geq \frac{\eta}{2} \sum_{k=1}^{K} \left\| \nabla F_0(\boldsymbol{\theta}_k^t) \right\|^2 , \tag{75}
$$

$$
= \frac{\eta}{2} \sum_{k=1}^{K} \left\| \nabla_{\boldsymbol{\theta}_k} \mathcal{L}\left( \boldsymbol{\Theta}^t \right) \right\|^2 . \tag{76}
$$

Then we have,

$$
\mathcal{L}(\boldsymbol{\Theta}^{T}) - \mathcal{L}(\boldsymbol{\Theta}^{0}) = \sum_{t=0}^{T-1} \mathcal{L}(\boldsymbol{\Theta}^{t+1}) - \mathcal{L}(\boldsymbol{\Theta}^{t}) , \tag{77}
$$

$$
\geq \frac{\eta}{2} \sum_{t=0}^{T-1} \sum_{k=1}^{K} \left\| \nabla_{\boldsymbol{\theta}_k} \mathcal{L}\left( \boldsymbol{\Theta}^t \right) \right\|^2 . \tag{78}
$$

Because

$$\frac{1}{\eta} = \frac{6L\sigma^2}{\sqrt{(40L+9\sigma^2)^2+96L\sigma^2}-(40L+9\sigma^2)} \,, \tag{79}$$

$$= \frac{6L\sigma^2\left(\sqrt{(40L+9\sigma^2)^2+96L\sigma^2}+(40L+9\sigma^2)\right)}{96L\sigma^2} \,, \tag{80}$$

$$= \frac{\sqrt{(40L+9\sigma^2)^2+96L\sigma^2}+(40L+9\sigma^2)}{16} \,, \tag{81}$$

$$\leq \frac{80L+18\sigma^2}{16} \,, \tag{82}$$

$$= \frac{40L+9\sigma^2}{8} \,. \tag{83}$$

That is,

$$\frac{1}{T}\sum_{t=0}^{T-1}\sum_{k=1}^{K}\left\|\nabla_{\boldsymbol{\theta}_k}\mathcal{L}\left(\boldsymbol{\Theta}^t\right)\right\|^2 \leq \frac{(40L+9\sigma^2)\left(\mathcal{L}(\boldsymbol{\Theta}^*)-\mathcal{L}(\boldsymbol{\Theta}^0)\right)}{4T} \,. \tag{84}$$

$\square$

## C  PROOF OF THE CONVERGENCE RATE OF FEDRC

**Assumption 3** (Unbiased gradients and bounded variance). *Each client* $i \in [M]$ *can sample a random batch* $\xi$ *from* $\mathcal{D}_i$ *and compute an unbiased estimator* $g_i(\xi, \boldsymbol{\theta})$ *of the local gradient with bounded variance, i.e.,* $\mathbb{E}_\xi[g_i(\xi,\boldsymbol{\theta})] = \frac{1}{N_i}\sum_{j=1}^{N_i}\nabla_{\boldsymbol{\theta}}f(\mathbf{x}_{i,j}, y_{i,j}, \boldsymbol{\theta})$ *and* $\mathbb{E}_\xi\left\|g_i(\xi,\boldsymbol{\theta})-\frac{1}{N_i}\sum_{i=1}^{N_i}\nabla_{\boldsymbol{\theta}}f(\mathbf{x}_{i,j}, y_{i,j}, \boldsymbol{\theta})\right\|^2 \leq \delta^2.$

**Assumption 4** (Bounded dissimilarity). *There exist* $\beta$ *and* $G$ *such that :*

$$\sum_{i=1}^{M}\frac{N_i}{N}\left\|\frac{1}{N_i}\sum_{j=1}^{N_i}\sum_{k=1}^{K}f(\mathbf{x}_{i,j}, y_{i,j}, \boldsymbol{\theta}_k)\right\|^2 \leq G^2+\beta^2\left\|\frac{1}{N}\sum_{i=1}^{M}\sum_{j=1}^{N_i}\sum_{k=1}^{K}f(\mathbf{x}_{i,j}, y_{i,j}, \boldsymbol{\theta}_k)\right\|^2 \,.$$

**Theorem C.1** (Convergence rate of FedRC). *Under Assumptions 1- 4, when clients use SGD as local solver with learning rate* $\eta \leq \frac{1}{\sqrt{T}}$, *and run FedRC for* $T$ *communication rounds, we have*

$$\frac{1}{T}\sum_{t=1}^{T}\mathbb{E}\left[\left\|\nabla_{\boldsymbol{\Theta}}\mathcal{L}(\boldsymbol{\Omega}^t, \boldsymbol{\Theta}^t)\right\|_F^2\right] \leq \mathcal{O}\left(\frac{1}{\sqrt{T}}\right) \,, \tag{85}$$

$$\frac{1}{T}\sum_{t=1}^{T}\mathbb{E}\left[\Delta_{\boldsymbol{\Omega}}\mathcal{L}(\boldsymbol{\Omega}^t, \boldsymbol{\Theta}^t)\right] \leq \mathcal{O}\left(\frac{1}{T^{\frac{3}{4}}}\right) \,, \tag{86}$$

*where* $\Delta_{\boldsymbol{\Omega}}\mathcal{L}(\boldsymbol{\Omega}^t, \boldsymbol{\Theta}^t) = \mathcal{L}(\boldsymbol{\Omega}^t, \boldsymbol{\Theta}^{t+1}) - \mathcal{L}(\boldsymbol{\Omega}^t, \boldsymbol{\Theta}^t).$

*Proof.* Constructing

$$g_i^t(\boldsymbol{\Omega}, \boldsymbol{\Theta}) = \frac{1}{N_i}\sum_{j=1}^{N_i}\sum_{k=1}^{K}\gamma_{i,j;k}^t\left(f(\mathbf{x}_{i,j}, y_{i,j}, \boldsymbol{\theta}_k)+\log(\mathcal{P}_k(y))-\log(\omega_{i;k})+\log(\gamma_{i,j;k}^t)\right) \,. \tag{87}$$

Then we would like to show (1) $g_i^t(\boldsymbol{\Omega}, \boldsymbol{\Theta})$ is L-smooth to $\boldsymbol{\Theta}$, (2) $g_i^t(\boldsymbol{\Omega}, \boldsymbol{\Theta}) \geq -\mathcal{L}_i(\boldsymbol{\Omega}, \boldsymbol{\Theta})$, (3) $g_i^t(\boldsymbol{\Omega}, \boldsymbol{\Theta})$ and $-\mathcal{L}_i(\boldsymbol{\Omega}, \boldsymbol{\Theta})$ have the same gradient on $\boldsymbol{\theta}$, and (4) $g_i^t(\boldsymbol{\Omega}^{t-1}, \boldsymbol{\Theta}^{t-1}) = \mathcal{L}_i(\boldsymbol{\Omega}^{t-1}, \boldsymbol{\Theta}^{t-1})$. When these conditions are satisfied, we can directly use Theorem 3.2$'$ of (Marfoq et al., 2021) to derive the final convergence rate. Firstly, it is obviously that $g_i^t(\boldsymbol{\Omega}, \boldsymbol{\Theta})$ is L-smooth to $\boldsymbol{\Theta}$ as it is a linear combination of $K$ smooth functions. Besides, we can easily obtain that

$$\frac{\partial g_i^t(\boldsymbol{\Omega}, \boldsymbol{\Theta})}{\partial \boldsymbol{\theta}_k} = \frac{1}{N_i}\sum_{j=1}^{N_i}\gamma_{i,j;k}^t\nabla_{\boldsymbol{\theta}_k}f(\mathbf{x}_{i,j}, y_{i,j}, \boldsymbol{\theta}_k) \,. \tag{88}$$

This is align with the gradient of $-\mathcal{L}_i(\mathbf{\Omega}, \mathbf{\Theta})$ as shown in Theorem A.1. Then define $r(\mathbf{\Omega}, \mathbf{\Theta}) = g_i^t(\mathbf{\Omega}, \mathbf{\Theta}) + \mathcal{L}_i(\mathbf{\Omega}, \mathbf{\Theta})$, we will have

$$r(\mathbf{\Omega}, \mathbf{\Theta}) = g_i^t(\mathbf{\Omega}, \mathbf{\Theta}) + \mathcal{L}_i(\mathbf{\Omega}, \mathbf{\Theta}) \tag{89}$$

$$= \frac{1}{N_i} \sum_{j=1}^{N_i} \sum_{k=1}^{K} \gamma_{i,j;k}^t \left( \log(\gamma_{i,j;k}^t) - \log(\omega_{i;k} \tilde{\mathcal{I}}(\mathbf{x}, y, \boldsymbol{\theta}_k)) \right) + \mathcal{L}_i(\mathbf{\Omega}, \mathbf{\Theta}) \tag{90}$$

$$= \frac{1}{N_i} \sum_{j=1}^{N_i} \sum_{k=1}^{K} \gamma_{i,j;k}^t \left( \log(\gamma_{i,j;k}^t) - \log(\omega_{i;k} \tilde{\mathcal{I}}(\mathbf{x}, y, \boldsymbol{\theta}_k)) + \log(\sum_{n=1}^{K} \omega_{i;n} \tilde{\mathcal{I}}(\mathbf{x}, y, \boldsymbol{\theta}_n)) \right) \tag{91}$$

$$= \frac{1}{N_i} \sum_{j=1}^{N_i} \sum_{k=1}^{K} \gamma_{i,j;k}^t \left( \log(\gamma_{i,j;k}^t) - \log\left( \frac{\omega_{i;k} \tilde{\mathcal{I}}(\mathbf{x}, y, \boldsymbol{\theta}_k)}{\sum_{n=1}^{K} \omega_{i;n} \tilde{\mathcal{I}}(\mathbf{x}, y, \boldsymbol{\theta}_n)} \right) \right) \tag{92}$$

$$= \frac{1}{N_i} \sum_{j=1}^{N_i} KL \left( \gamma_{i,j;k}^t \| \frac{\omega_{i;k} \tilde{\mathcal{I}}(\mathbf{x}, y, \boldsymbol{\theta}_k)}{\sum_{n=1}^{K} \omega_{i;n} \tilde{\mathcal{I}}(\mathbf{x}, y, \boldsymbol{\theta}_n)} \right) \geq 0. \tag{93}$$

Besides, from Equation 4, we can found that

$$\gamma_{i,j;k}^t = \frac{\omega_{i;k}^{t-1} \tilde{\mathcal{I}}(\mathbf{x}, y, \boldsymbol{\theta}_k^{t-1})}{\sum_{n=1}^{K} \omega_{i;n}^{t-1} \tilde{\mathcal{I}}(\mathbf{x}, y, \boldsymbol{\theta}_n^{t-1})}, \tag{94}$$

then the last condition is also satisfied. Therefore, we have shown that FedRC is a special case of the Federated Surrogate Optimization defined in (Marfoq et al., 2021), and the convergence rate is obtained by

$$\frac{1}{T} \sum_{t=1}^{T} \mathbb{E} \left[ \left\| \nabla_{\mathbf{\Theta}} \mathcal{L}(\mathbf{\Omega}^t, \mathbf{\Theta}^t) \right\|_F^2 \right] \leq \mathcal{O} \left( \frac{1}{\sqrt{T}} \right), \tag{95}$$

$$\frac{1}{T} \sum_{t=1}^{T} \mathbb{E} \left[ \Delta_{\mathbf{\Omega}} \mathcal{L}(\mathbf{\Omega}^t, \mathbf{\Theta}^t) \right] \leq \mathcal{O} \left( \frac{1}{T^{\frac{3}{4}}} \right). \tag{96}$$

$\square$

## D  RELATED WORKS

**Federated Learning with label distribution shifts.**   As the de facto FL algorithm, (McMahan et al., 2016; Lin et al., 2020) proposes to use local SGD steps to alleviate the communication bottleneck. However, the non-iid nature of local distribution hinders the performance of FL algorithms (Li et al., 2018; Wang et al., 2020b; Karimireddy et al., 2020b;a; Guo et al., 2021; Jiang & Lin, 2023). Therefore, designing algorithms to deal with the distribution shifts over clients is a key problem in FL (Kairouz et al., 2021). Most existing works only consider the label distribution skew among clients. Some techniques address local distribution shifts by training robust global models (Li et al., 2018; 2021), while others use variance reduction methods (Karimireddy et al., 2020b;a). However, the proposed methods cannot be used directly for concept shift because the decision boundary changes. Combining FedRC with other methods that address label distribution shifts may be an interesting future direction, but it is orthogonal to our approach in this work.

**Federated Learning with feature distribution shifts.**   Studies about feature distribution skew in FL mostly focus on domain generalization (DG) problem that aims to train robust models that can generalize to unseen feature distributions. (Reisizadeh et al., 2020) investigates a special case that the local distribution is perturbed by an affine function, i.e. from $x$ to $Ax + b$. Many studies focus on adapting DG algorithms for FL scenarios. For example, combining FL with Distribution Robust Optimization (DRO), resulting in robust models that perform well on all clients (Mohri et al., 2019; Deng et al., 2021); combining FL with techniques that learn domain invariant features (Peng et al., 2019; Wang et al., 2022a; Shen et al., 2021; Sun et al., 2022; Gan et al., 2021) to improve the generalization ability of trained models. All of the above methods aim to train a single robust feature extractor that can generalize well on unseen distributions. However, using a single model cannot solve the diverse distribution shift challenge, as the decision boundary changes when concept shifts occur.

**Federated Learning with concept shifts.** Concept shift is not a well-studied problem in FL. However, some special cases become an emerging topic recently. For example, research has shown that label noise can negatively impact model performance (Ke et al., 2022). Additionally, methods have been proposed to correct labels that have been corrupted by noise (Fang & Ye, 2022; Xu et al., 2022). Recently, (Jothimurugesan et al., 2022) also investigate the concept shift problem under the assumption that clients do not have concept shifts at the beginning of the training. However, it is difficult to ensure the non-existence of concept shifts when information about local distributions is not available in practice. Additionally, (Jothimurugesan et al., 2022) did not address the issue of label and feature distribution skew. In this work, we consider a more realistic but more challenging scenario where clients could have all three kinds of shifts with each other, unlike (Jothimurugesan et al., 2022) that needs to restrict the non-occurrence of concept shift at the initial training phase.

**Clustered Federated Learning.** Clustered Federated Learning (clustered FL) is a technique that groups clients into clusters based on their local data distribution to address the distribution shift problem. Various methods have been proposed for clustering clients, including using local loss values (Ghosh et al., 2020), communication time/local calculation time (Wang et al., 2022b), fuzzy $c$-Means (Stallmann & Wilbik, 2022), and hierarchical clustering (Zhao et al., 2020; Briggs et al., 2020; Sattler et al., 2020a). Some approaches, such as FedSoft (Ruan & Joe-Wong, 2022), combine clustered FL with personalized FL by using a soft cluster mechanism that allows clients to contribute to multiple clusters. Other methods, such as FeSEM (Long et al., 2023) and FedEM (Marfoq et al., 2021), employ an Expectation-Maximization (EM) approach and utilize the log-likelihood objective function, as in traditional EM. Our proposed FedRC overcomes the pitfalls of other clustered FL methods, and shows strong empirical effectiveness over other clustered FL methods. It is noteworthy that FedRC could further complement other clustered FL methods for larger performance gain.

# E  DISCUSSION: ADDING STOCHASTIC NOISE ON $\gamma_{i,j;k}$ PRESERVES CLIENT PRIVACY.

The approximation of $\mathcal{P}(y; \boldsymbol{\theta}_k)$ in (3) needs to gather $\sum_{i=1}^{M} \sum_{j=1}^{N_i} \mathbf{1}_{\{y_{i,j}=y\}} \gamma_{i,j;k}$, which may expose the local label distributions. To address this issue, additional zero-mean Gaussian noise can be added: $C_{y,i} = c_{\xi_i} + \sum_{j=1}^{N_i} \mathbf{1}_{\{y_{i,j}=y\}} \gamma_{i,j;k}$, where $c_{\xi_i} \sim \mathcal{N}\left(0, \xi_i^2\right)$, and $\xi_i$ is the standard deviation of the Gaussian noise for client $i$ (which can be chosen by clients). By definition of $C_y = 1/N \sum_{i=1}^{M} C_{y,i}$, we have

$$C_y := c_{\xi_g} + \frac{1}{N} \sum_{i=1}^{M} \sum_{j=1}^{N_i} \left( \mathbf{1}_{\{y_{i,j}=y\}} \gamma_{i,j;k} \right) , \tag{97}$$

where $c_{\xi_g} \sim \mathcal{N}\left(0, \sum_{i=1}^{M} \xi_i^2/N^2\right)$, which standard deviation decreases as $N$ increases. Therefore, when $N$ is large, we can obtain a relatively precise $C_y \approx \frac{1}{N} \sum_{i=1}^{M} \sum_{j=1}^{N_i} \mathbf{1}_{\{y_{i,j}=y\}} \gamma_{i,j;k}$ without accessing the true $\sum_{j=1}^{N_i} \mathbf{1}_{\{y_{i,j}=y\}} \gamma_{i,j;k}$. Note that the performance of FedRC will not be significantly affected by the magnitude of noise, as justified in Figure 14(a) and Table 10.

# F  DISCUSSION: ADAPTIVE FEDRC FOR DECIDING THE NUMBER OF CONCEPTS

In previous sections, the algorithm design assumes a fixed number of clusters equivalent to the number of concepts. However, in real-world scenarios, the number of concepts may be unknown. This section explores how to determine the number of clusters adaptively in such cases.

**Proposal: extending FedRC to determine the number of concepts.** FedRC effectively avoids "bad clustering" dominated by label skew and feature skew, i.e., variations in $\mathcal{P}(y; \boldsymbol{\theta}_k)$ or $\mathcal{P}(\mathbf{x}; \boldsymbol{\theta}_k)$ among clusters. For example, in the case of data $(\mathbf{x}_1, y_1)$ that $\mathcal{P}(y_1; \boldsymbol{\theta}_k)$ or $\mathcal{P}(\mathbf{x}_1; \boldsymbol{\theta}_k)$ is small, the data will assign larger weights to model $\boldsymbol{\theta}_k$ in the next E-step (i.e. (4)) of FedRC. It would increase the values of $\mathcal{P}(y_1; \boldsymbol{\theta}_k)$ and $\mathcal{P}(\mathbf{x}_1; \boldsymbol{\theta}_k)$ to avoid the unbalanced label or feature distribution within each cluster.

As a result, data only have feature or label distribution shifts will not be assigned to different clusters, and empty clusters can be removed: we conjecture that when (1) the weights $\gamma_{ijk}$ converge and (2) $K$ is larger than the number of concepts, we can determine the number of concepts by removing model $\boldsymbol{\theta}_k$ when $\sum_{i,j} \gamma_{i,j;k} \to 0$. More discussions in Appendix F.

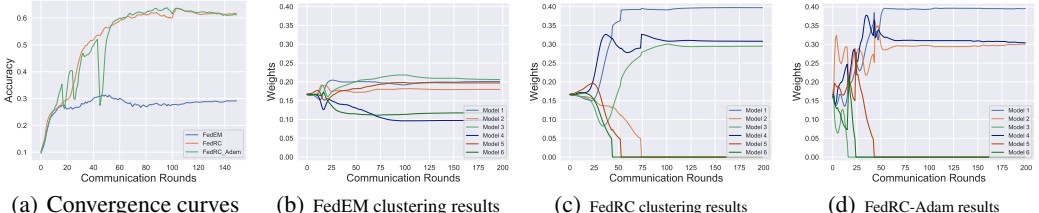

(a) Convergence curves    (b) FedEM clustering results    (c) FedRC clustering results    (d) FedRC-Adam results

Figure 7: **Adaptive process of FedRC.** We split CIFAR10 dataset into 300 clients, initialize 6 clusters, and report the global accuracy per round for the convergence result. We also show the clustering results by calculating weights by $\sum_{i,j} \gamma_{i,j;k} / \sum_{i,j,k} \gamma_{i,j;k}$, representing the proportion of clients that select model $k$. We use $\delta = 0.05$ here. Ablation study on $\delta$ refers to Appendix I.4.

**Verifying the proposal via empirical experiments.** We conduct experiments on CIFAR10 using the same settings as in Section 3 and run FedRC with $K = 4$ (greater than the concept number 3). The clustering results of FedRC are shown in Figures 4(c)- 4(d). The results indicate that data with concept shifts are assigned to clusters $2, 3, 4$, while data without concept shifts are not divided into different clusters, leading to minimal data in cluster 1. Hence, cluster 1 can be removed.

**Design of adaptive FedRC.** The process of adaptive FedRC involves initially setting a large number of clusters $K$ and checking if any model $\boldsymbol{\theta}_k$ meets the following condition when the weights converge: $\frac{1}{N} \sum_{i=1}^{M} \sum_{j=1}^{N_i} \gamma_{i,j;k} < \delta$. The value of $\delta$ sets the threshold for model removal. For more details regarding the model removal, refer to Appendix H.4.

### F.1 Results of Adaptive FedRC

In this section, we set the number of clusters $K = 6$, which is greater than the number of concepts, to demonstrate the effectiveness of the FedRC on automatically deciding the number of concepts.

**Accelerating the convergence of $\gamma_{i,j;k}$ by Adam.** The adaptive FedRC needs a full convergence of $\gamma_{i,j;k}$ before removing models, emphasizing the need to accelerate the convergence of $\gamma_{i,j;k}$. One solution is incorporating Adam (Kingma & Ba, 2014) into the optimization of $\gamma_{i,j;k}$, by treating $\gamma_{i,j;k}^{t+1} - \gamma_{i,j;k}^{t}$ as the gradient of $\gamma_{i,j;k}$ at round $t$. Please refer to Appendix H.3 for an enhanced optimization on $\gamma_{i,j;k}$. Figure 7 shows that FedRC exhibits faster convergence after introducing Adam.

**Effectiveness of FedRC in determining the number of concepts.** In Figure 7, we show 1) FedRC can correctly find the number of concepts; 2) weights $\gamma_{ijk}$ in FedRC-Adam converge significantly faster than FedRC (from 75 to 40 communication rounds); 3) FedEM failed to decide the number of concepts, and the performance is significantly worse than FedRC.

## G Discussion: Interpretation of the objective function

In this section, we would like to give a more clear motivation about why maximizing our objective function defined by Equation (1) can achieve *principles of robust clustering*. In Figure 8, we construct a toy case to show: (1) The value of $\mathcal{L}(\boldsymbol{\Omega}, \boldsymbol{\Theta})$ decreases when the data has concept shifts with the distribution of cluster $k$ is assigned to the cluster $k$. (2) The value of $\mathcal{L}(\boldsymbol{\Omega}, \boldsymbol{\Theta})$ is not sensitive to the feature or label distribution shifts.

Furthermore, in Figure 9, we demonstrate that RobustCluster can solve *principles of robust clustering* by assigning data with concept shifts to different clusters while maintaining data without concept shifts in the same clusters.

## H Experiment Details

### H.1 Algorithm Framework of FedRC

In Algorithm 2, we summarize the whole process of FedRC. In detail, at the beginning of round $t$, the server transmits parameters $\boldsymbol{\Theta}^t = [\boldsymbol{\theta}_1^t, \cdots, \boldsymbol{\theta}_K^t]$ to clients. Clients then locally update $\gamma_{i,j;k}^{t+1}$ and $\omega_{i;k}^{t+1}$ using (4) and (5), respectively. Then clients initialize $\boldsymbol{\theta}_{i,k}^{t,0} = \boldsymbol{\theta}_k^t$, and update parameters $\boldsymbol{\theta}_{i,k}^{t,\tau}$ for $\mathcal{T}$ local steps:

$$\boldsymbol{\theta}_{i,k}^{t,\tau} = \boldsymbol{\theta}_{i,k}^{t,\tau-1} - \frac{\eta_l}{N_i} \sum_{j=1}^{N_i} \gamma_{i,j;k}^t \nabla_{\boldsymbol{\theta}} f_{i,k}(\mathbf{x}_{i,j}, y_{i,j}, \boldsymbol{\theta}_{i,k}^{t,\tau-1}), \tag{98}$$

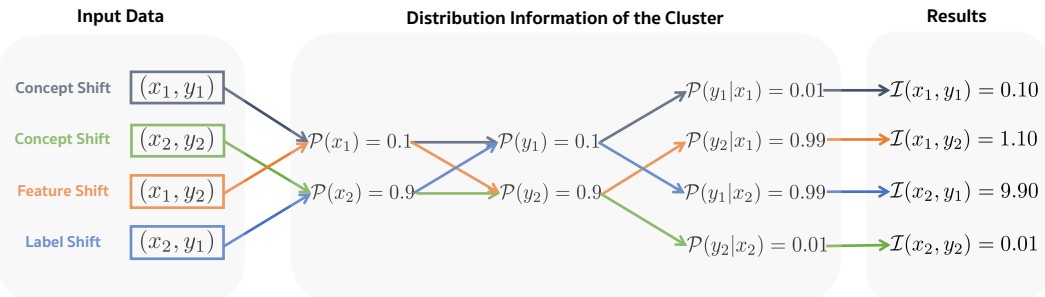

Figure 8: **A toy case to illustrate the change of objective functions with different distribution shifts.** We created a simple example to demonstrate how the value of $\mathcal{L}(\mathbf{x}, y, \boldsymbol{\theta}_k)$ changes as the type of distribution shifts changes. Note that larger $\mathcal{I}(\mathbf{x}, y)$ indicates larger $\mathcal{L}(\boldsymbol{\Omega}, \boldsymbol{\Theta})$.

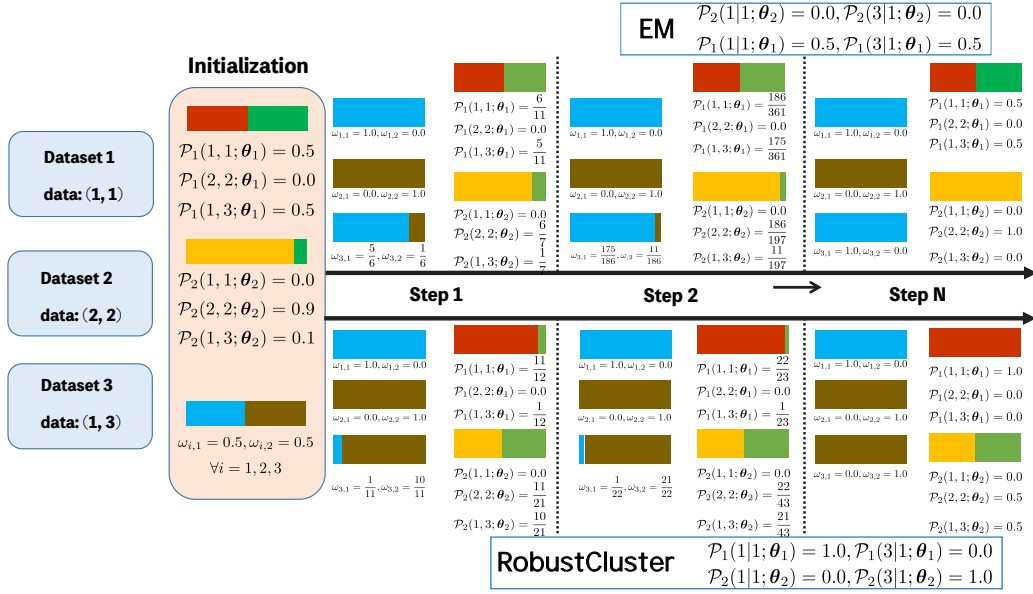

Figure 9: **Toy case compare RobustCluster and EM algorithms.** We create a simple example with three datasets that have an equal number of data points and compare results of EM algorithm and RobustCluster. We show $\omega_{i;k}$ and $\mathcal{P}(\mathbf{x}, y; \boldsymbol{\theta}_k)$ at each step, as well as the conditional distribution $\mathcal{P}(y|\mathbf{x})$ for classification tasks. EM algorithm has low classification results on dataset 1 and 3, but RobustCluster always finds a model $\boldsymbol{\theta}_k$ where $\mathcal{P}(y|\mathbf{x}; \boldsymbol{\theta}_k) = 1$ for all data in all datasets.

where $\tau$ is the current local iteration, and $\eta_l$ is the local learning rate. In our algorithm, the global aggregation step uses the FedAvg method as the default option. However, other global aggregation methods e.g. (Wang et al., 2020a;b) can be implemented if desired. We summarize the optimization of FedRC in Figure 10, and include more details about the model prediction in Appendix H.2. We also include the discussion about the convergence rate of FedRC in Appendix C.

## H.2 MODEL PREDICTION OF FEDRC FOR SUPERVISED TASKS.

The model prediction of FedRC for supervised tasks is performed using the same method as in FedEM (Marfoq et al., 2021). I.e., given data $\mathbf{x}$ for client $i$, the predicted label is calculated as $y_{\text{pred}} = \sum_{k=1}^K \omega_{i;k} \sigma(m_{i,k}(\mathbf{x}, \boldsymbol{\theta}_k))$, where $m_{i,k}(\mathbf{x}, \boldsymbol{\theta}_k)$ is the output of model $\boldsymbol{\theta}_k$ given data $\mathbf{x}$, and $\sigma$ is the softmax function. Note that the aforementioned $f_{i;k}(\mathbf{x}, y; \boldsymbol{\theta}_k)$ is the cross entropy loss of $\sigma(m_{i,k}(\mathbf{x}, \boldsymbol{\theta}_k))$ and $y$. When new clients join the system, they must first perform the E-step to obtain the optimal weights $\gamma_{i,j;k}$ and $\omega_{i;k}$ by (4) on their local train (or validation) datasets before proceeding to test.

---

**Algorithm 2** FedRC Algorithm Framework

---

**Require:** The number of models $K$, initial parameters $\Theta^0 = [\boldsymbol{\theta}_1^0, \cdots, \boldsymbol{\theta}_K^0]$, global learning rate $\eta_g$, initial weights $\omega_{i;k}^0 = \frac{1}{K}$ for any $i, k$, and $\gamma_{i,j;k} = \frac{1}{K}$ for any $i, j, k$, local learning rate $\eta_l$.
**Ensure:** Trained parameters $\Theta^T = [\boldsymbol{\theta}_1^T, \cdots, \boldsymbol{\theta}_K^T]$ and weights $\omega_{i;k}^T$ for any $i, k$.

1: **for** round $t = 1, \ldots, T$ **do**
2:     (Optional) **Adapt:** Check & remove model via Algorithm 3.
3:     **Communicate** $\Theta^t$ to the chosen clients.
4:     **for client** $i \in \mathcal{S}^t$ **in parallel do**
5:         Initialize local model $\boldsymbol{\theta}_{i,k}^{t,0} = \boldsymbol{\theta}_k^t, \forall k$.
6:         Update $\gamma_{i,j;k}^t, \omega_{i;k}^t$ by (4), $\forall j, k$.
7:         Local update $\boldsymbol{\theta}_{i,k}^{t,\mathcal{T}}$ by (6) , $\forall k$.
8:         **Communicate** $\Delta\boldsymbol{\theta}_{i,k}^t \leftarrow \boldsymbol{\theta}_{i,k}^{t,\mathcal{T}} - \boldsymbol{\theta}_{i,k}^{t,0}, \forall k$.
9:     $\boldsymbol{\theta}_k^{t+1} \leftarrow \boldsymbol{\theta}_k^t + \frac{\eta_g}{\sum_{i \in \mathcal{S}^t} N_i} \sum_{i \in \mathcal{S}^t} N_i \Delta\boldsymbol{\theta}_{i,k}^t, \forall k$.

---

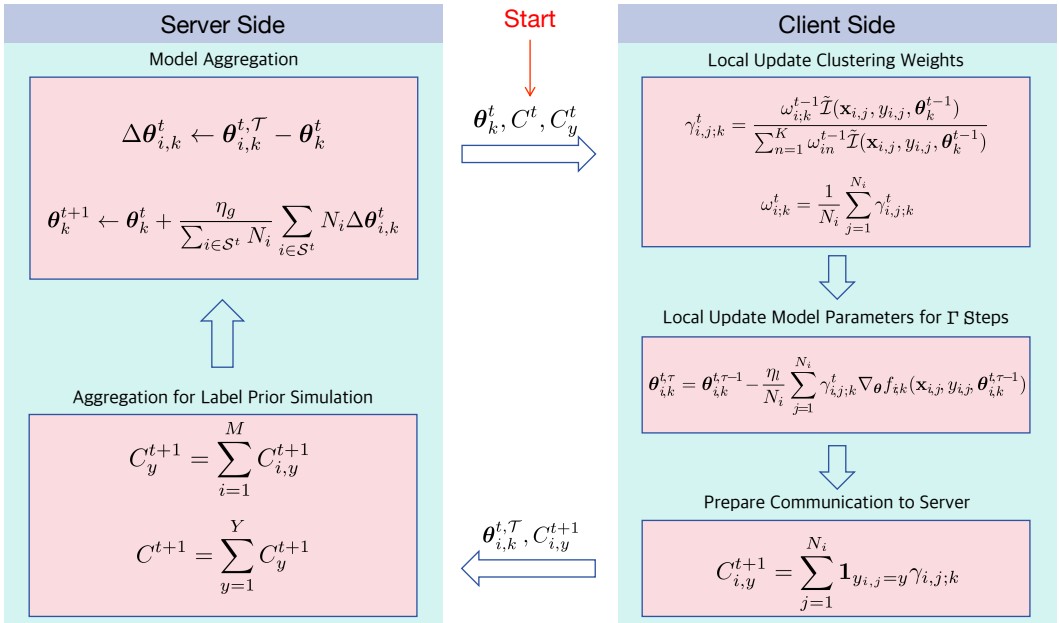

Figure 10: **Illustration of the optimization process of FedRC.**

---

**Algorithm 3** Check and remove model in FedRC

---

**Require:** Threshold value $\delta$, number of clients $M$, number of data in each client $N_i$, number of model $K$, $\gamma_k = \frac{1}{N} \sum_{i=1}^{M} \sum_{j=1}^{N_i} \gamma_{i,j;k}$ for each $k$.

1: **for** model $k = 1, \ldots, K$ **do**
2:    **if** $\frac{1}{N} \sum_{i=1}^{M} \sum_{j=1}^{N_i} \gamma_{i,j;k} < \delta$ **then**
3:      Remove model $k$.
4:      **for** client $i = 1, \ldots, M$ **do**
5:        Remove $\gamma_{i,j;k}$ for all $j$.
6:        **for** client $\tau = 1, \ldots, K, \tau \neq k$ **do**
7:          $\gamma_{i,j;\tau} = \frac{\gamma_{i,j;\tau}}{\sum_{\tau'} \gamma_{i,j;\tau'}}$ for all $j$.
8:        Remove $\omega_{i;k}$.
9:        Update $\omega_{i;\tau}$ by Equation (4) for all $\tau \neq k$.

---

### H.3 OPTIMIZING $\gamma_{i,j;k}$ FOR FEDRC-ADAM

We summarize the optimization steps of FedRC-Adam on $\gamma_{i,j;k}$ as follows:

$$\tilde{\gamma}_{i,j;k}^t = \frac{\omega_{i;k}^{t-1} \tilde{\mathcal{I}}(\mathbf{x}_{i,j}, y_{i,j}, \boldsymbol{\theta}_k)}{\sum_{n=1}^{K} \omega_{i;n}^{t-1} \tilde{\mathcal{I}}(\mathbf{x}_{i,j}, y_{i,j}, \boldsymbol{\theta}_k)}, \tag{99}$$

$$g_{i,j;k} = \gamma_{i,j;k}^{t-1} - \tilde{\gamma}_{i,j;k}^t, \tag{100}$$

$$\nu_{i,j;k}^t = (1 - \beta_1) g_{i,j;k} + \beta_1 \nu_{i,j;k}^{t-1}, \tag{101}$$

$$a_{i,j;k}^t = (1 - \beta_2) g_{i,j;k}^2 + \beta_2 a_{i,j;k}^{t-1}, \tag{102}$$

$$\gamma_{i,j;k}^t = \gamma_{i,j;k}^{t-1} - \alpha \frac{\nu_{i,j;k}^t / (1 - \beta_1)}{\sqrt{a_{i,j;k}^t / (1 - \beta_2)} + \epsilon}, \tag{103}$$

$$\gamma_{i,j;k}^t = \frac{max(\gamma_{i,j;k}^t, 0)}{\sum_{n=1}^{K} max(\gamma_{i,j;n}^t, 0)}, \tag{104}$$

where we set $\alpha = 1.0$, $\beta_1 = 0.9$, $\beta_2 = 0.99$, and $\epsilon = 1e-8$ in practice by default. The Equation (104) is to avoid $\gamma_{i,j;k}^t < 0$ after adding the momentum.

### H.4 REMOVING THE MODELS IN ADAPTIVE PROCESS

In Algorithm 3, we show how to remove the models once we decide the model could be removed. Once the model $\boldsymbol{\theta}_k$ is decided to be removed, the server will broadcast to all the clients, and clients will remove the local models, and normalize $\gamma_{i,j;k}$ and $\omega_{i;k}$ by Line 7 and Line 9 in Algorithm 3.

### H.5 FRAMEWORK AND BASELINE ALGORITHMS

We extend the public code provided by (Marfoq et al., 2021) in this work. For personalized algorithms that don't have a global model, we average the local models to create a global model and evaluate it on global test datasets. When testing non-participating clients on clustered FL algorithms (IFCA, CFL, and FeSEM), we assign them to the cluster they performed best on during training.

### H.6 EXPERIMENTAL ENVIRONMENT

For all experiments, we use NVIDIA GeForce RTX 3090 GPUs. Each simulation trail with 200 communication rounds and 3 clusters takes about 9 hours.

### H.7 MODELS AND HYPER-PARAMETER SETTINGS

We use a three-layer CNN for the FashionMNIST dataset, and use pre-trained MobileNetV2 (Sandler et al., 2018) for CIFAR10 and CIFAR100 datasets. We set the batch size to 128, and run 1 local epoch in each communication round by default. We use SGD optimizer and set the momentum to 0.9. The learning rates are chosen in $[0.01, 0.03, 0.06, 0.1]$, and we run each algorithm for 200 communication rounds, and report the best result of each algorithm. We also include the results of CIFAR10 and

CIFAR100 datasets using ResNet18 (He et al., 2016) in Table 2. For each clustered FL algorithm (including FedRC), we initialize 3 models by default following the ideal case. We also investigate the adaptive process by initialize 6 models at the beginning in Figure 7.

### H.8   DATASETS CONSTRUCTION

**FashionMNIST dataset.**   We split the FashionMNIST dataset into 300 groups of clients using LDA with a value of 1.0 for alpha. 30% of the clients will not have any changes to their images or labels. For 20% of the clients, labels will not be changed, and we will add synthetic corruptions following the idea of FashionMNIST-C with a random level of severity from 1 to 5. 25% of the clients will have their labels changed to $C - 1 - y$, where $C$ is the number of classes, and $y$ is the true label. We will also add synthetic corruption to 20% of these clients (5% of 300 clients). Finally, 25% of the clients will have their labels changed to $(y + 1) \mod C$, and we will also add synthetic corruptions to 20% of these clients.

**CIFAR10 and CIFAR100 datasets.**   For the MobileNetV2 model, we divided the dataset into 300 smaller groups of data (called "clients"). To ensure that each group had enough data to be well-trained, we first made three copies of the original dataset. Then, we used a technique called Latent Dirichlet Allocation (LDA) with a parameter of $\alpha = 1.0$ to divide the dataset into the 300 clients. For the ResNet18 model, we also used LDA with $\alpha = 1.0$, but we divided the dataset into 100 clients without making any copies. Then 30% of the clients will not have any changes to their images or labels. For 20% of the clients, labels will not be changed, and we will add synthetic corruptions following the idea of FashionMNIST-C with a random level of severity from 1 to 5. 25% of the clients will have their labels changed to $C - 1 - y$, where $C$ is the number of classes, and $y$ is the true label. We will also add synthetic corruption to 20% of these clients (5% of 300 clients). Finally, 25% of the clients will have their labels changed to $(y + 1) \mod C$, and we will also add synthetic corruptions to 20% of these clients.

**Test clients construction.**   We directly use the test datasets provided by FashionMNIST, CIFAR10, and CIFAR100 datasets, and construct 3 test clients. The labels for the first client remain unchanged. For the second client, the labels are changed to $C - 1 - y$, and for the third client, the labels are changed to $(y + 1) \mod C$.

## I   ADDITIONAL EXPERIMENT RESULTS

### I.1   RESULTS ON REAL-WORLD CONCEPT SHIFT DATASETS

In this section, we include two real-world datasets that naturally have concept shifts, including Airline and Electricity datasets, and the prepossess we used is the same as (Tahmasbi et al., 2021):

- **Airline (Ikonomovska, 2020):** The real-world dataset used in this study comprises flight schedule records and binary class labels indicating whether a flight was delayed or not. Concept drift may arise due to changes in flight schedules, such as alterations in the day, time, or duration of flights. For our experiments, we utilized the initial 58100 data points of the dataset. The dataset includes 13 features.
- **Electricity (Harries et al., 1999):** The real-world dataset utilized in this study includes records from the New South Wales Electricity Market in Australia, with binary class labels indicating whether the price changed (i.e., up or down). Concept drift in this dataset may arise from shifts in consumption patterns or unforeseen events.

For each dataset, we first construct two test datasets that have concept shifts with each other, and have balanced label distribution. For the remaining data, we split the dataset to 300 clients using LDA with $\alpha = 0.1$. We use a three layer MLP for training, and initialize 3 models for all the algorithms. We set the batch-size to 128, learning rate to 0.01, and run algorithms for 500 communication rounds. Results in Table 3 show FedRC outperform other clustered FL algorithms significantly on these two datasets.

### I.2   VISUALIZATION OF CLUSTER DIVISION RESULTS

In this section, we provide the comprehensive visualization of the cluster division results of FedRC. The experiment settings are the same to Section 5.1.

**Cluster division results of baseline algorithms on CIFAR10.**   In Figure 12 and Figure 11, we present the cluster division results of both existing methods and FedRC on CIFAR-10 datasets.

Table 3: **Results of algorithms on Airline and Electricity datasets.** We report the best mean accuracy on test datasets.

| Algorithms | Airline | Electricity |
|---|---|---|
| FedAvg | 51.69 | 48.83 |
| FeSEM | 51.25 | 50.21 |
| IFCA | 50.14 | 49.67 |
| FedEM | 50.36 | 54.79 |
| FedRC | **59.14** | **60.92** |

Figure 11: **Clustering results of FedRC w.r.t. classes/feature styles/concepts.** A larger circle size indicates that more data points associated with a class, feature style, or concept are assigned to cluster $k$. The number of clusters is selected from the range $[3, 4]$, and the number of concepts is consistently set to 3 for all experiments.

The results reveal that: (1) existing methods tend to assign data samples with the same labels to the same clusters (refer to Figures 12(a), 12(d), 12(g)); (2) existing methods are ineffective in assigning data samples with concept shifts to different clusters; instead, samples with the same concepts are uniformly distributed across various clusters (refer to Figures 12(c), 12(f), 12(i), 12(l)); and (3) the FedRC correctly assign samples with concept shifts into different clusters (refer to Figures 11(c), 11(f)).

**The effectiveness of FedRC remains on various datasets.** In Figure 13, we present the cluster division results of FedRC for the CIFAR10, CIFAR100, and Tiny-ImageNet datasets. The findings demonstrate that FedRC consistently assigns data samples with concept shifts to distinct clusters.

## I.3 ABLATION STUDY ON FEDRC

In this section, we present additional findings from our ablation study on local steps, concept count, and client involvement.

**Results with additional PFL baselines.** In Table 4, we include more personalized FL methods, including local, pFedMe (T Dinh et al., 2020), and APFL (Deng et al., 2020). Results show personalized FL methods struggle to generalize to global distributions, in contrast to the FedRC-FT constructed by fine-tuning FedRC for one local epoch can achieve comparable performance with personalized FL methods on local accuracy while maintaining the high global performance.

**Ablation studies on local steps.** In Table 5, we demonstrate the impact of the number of local epochs on the performance of different algorithms. Our results indicate that FedRC consistently

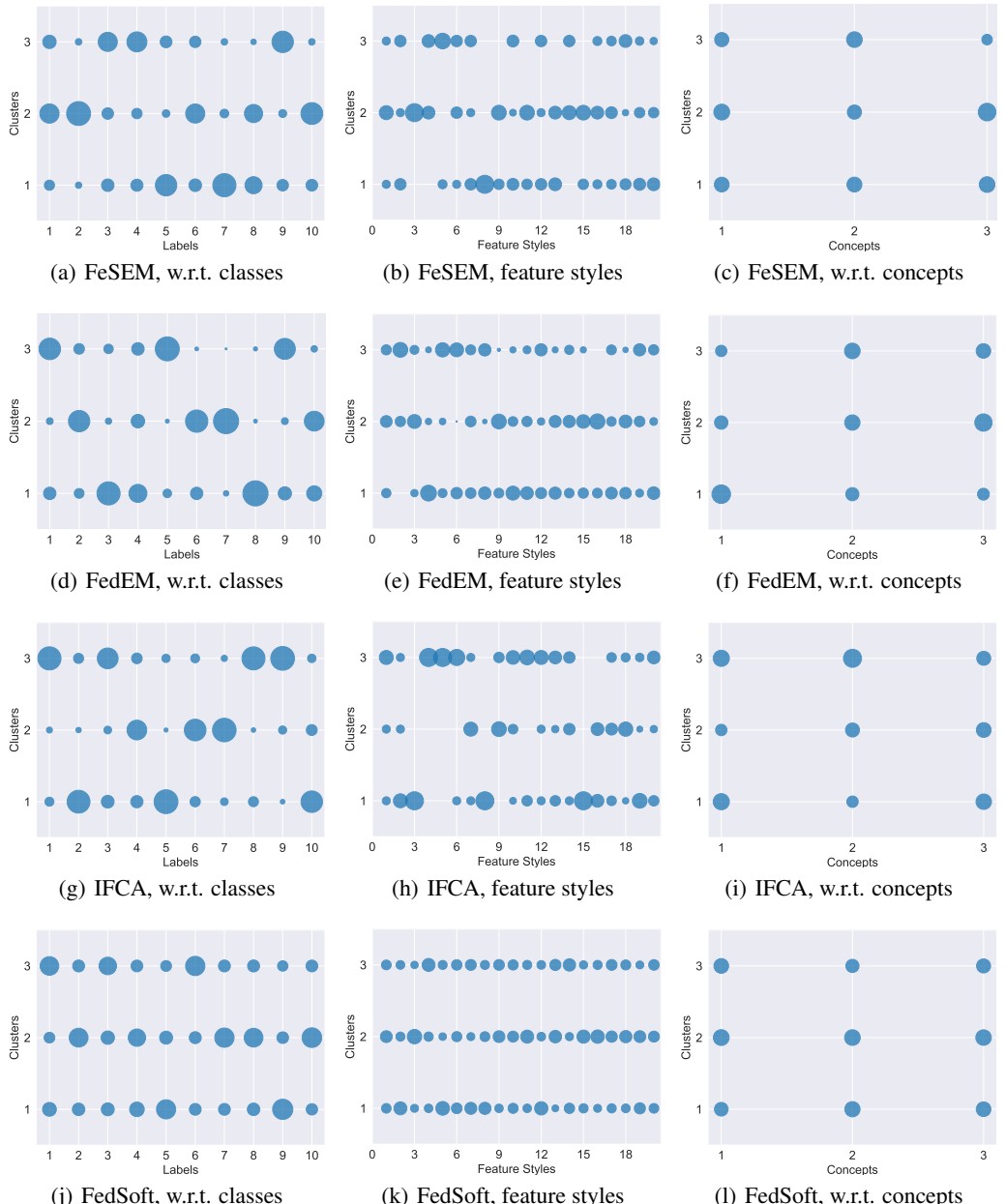

Figure 12: **Clustering results w.r.t. classes/feature styles/concepts.** After data construction, each data $\mathbf{x}$ will have a class $y$, feature style $f$, and concept $c$. We report the percentage of data points associated with a class, feature style, or concept are assigned to cluster k. For example, for a circle centered at position $(y, k)$, a larger circle size signifies that more data points with class $y$ are assigned to cluster $k$.

outperforms the other baseline algorithms, even as the number of local epochs increases. However, we also observe that FedRC is relatively sensitive to the number of local epochs. This may be due to the fact that a large number of local steps can cause the parameters to drift away from the true global optima, as previously reported in several FL studies (e.g. (Karimireddy et al., 2020b; Li et al., 2020; 2021)). In the E step, this can make it more difficult for the algorithm to accurately find $\gamma_{i,j;k}$ based on sub-optimal parameters.

**Ablation studies on partial client participation.** We have included the ablation study about the number of clients participating in each round in the main paper. In Table 6, we report the final accuracies on both local and global test datasets to include more details. Results show FedRC is robust to the partial client participation.

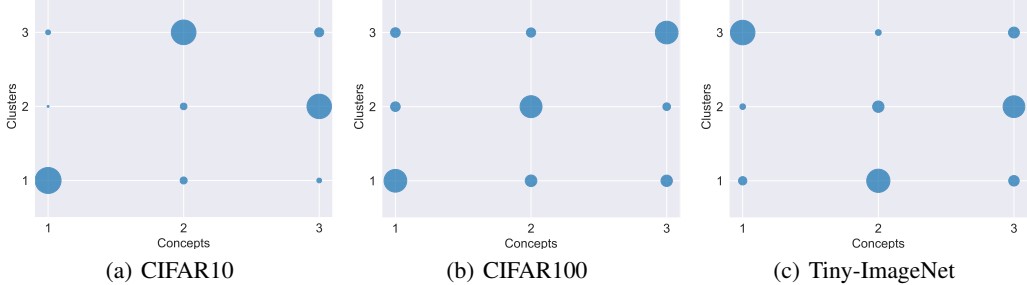

|  | (a) CIFAR10 | (b) CIFAR100 | (c) Tiny-ImageNet |

Figure 13: **Clustering results of FedRC w.r.t. concepts on various datasets.** After data construction, each data **x** will have a class $y$, feature style $f$, and concept $c$. We report the percentage of data points associated with the concepts that are assigned to cluster k. For example, for a circle centered at position $(c, k)$, a larger circle size signifies that more data points with concept $c$ are assigned to cluster $k$.

Table 4: **Performance of algorithms over various datasets and neural architectures.** We evaluated the performance of our algorithms using the FashionMNIST, CIFAR10 and CIFAR100 datasets split into 300 clients. We initialized 3 models for clustered FL methods and reported mean local and global test accuracy on the round that achieved the best train accuracy for each algorithm. We report FedRC-FT by fine-tuning FedRC for one local epoch; "CFL (3)" refers to restricting the number of models in CFL (Sattler et al., 2020b) to 3. We highlight the best and the second best results for each of the two main blocks, using **bold font** and blue text.

| Algorithm | FashionMNIST (CNN) | | CIFAR10 (MobileNetV2) | | CIFAR100 (MobileNetV2) | |
|---|---|---|---|---|---|---|
|  | Local | Global | Local | Global | Local | Global |
| FedAvg | 42.12 ±0.33 | 34.35 ±0.92 | 30.28 ±0.38 | 30.47 ±0.76 | 12.72 ±0.82 | 10.53 ±0.57 |
| IFCA | 47.90 ±0.60 | 31.30 ±2.69 | 43.76 ±0.40 | 26.62 ±3.34 | 17.46 ±0.10 | 9.12 ±0.78 |
| CFL (3) | 41.77 ±0.40 | 33.53 ±0.35 | 41.49 ±0.64 | 29.12 ±0.02 | **26.36** ±0.33 | 7.15 ±1.10 |
| FeSEM | 60.99 ±1.01 | 47.63 ±0.99 | 45.32 ±0.16 | 30.79 ±0.02 | 18.46 ±3.96 | 9.76 ±0.64 |
| FedEM | 56.64 ±2.14 | 28.08 ±0.92 | 51.31 ±0.97 | 43.35 ±2.29 | 17.95 ±0.08 | 9.72 ±0.22 |
| FedRC | **66.51** ±2.39 | **59.00** ±4.91 | **62.74** ±2.37 | **63.83** ±2.26 | 21.64 ±0.33 | **18.72** ±1.90 |
| local | 92.79 ±0.58 | 12.92 ±4.93 | 82.71 ±0.25 | 10.59 ±0.69 | 86.58 ±0.08 | 1.00 ±0.18 |
| pFedMe | 93.13 ±0.32 | 14.77 ±3.39 | 82.63 ±0.05 | 9.87 ±0.85 | 86.83 ±0.23 | 1.20 ±0.04 |
| APFL | **93.29** ±0.16 | 33.40 ±0.89 | **85.03** ±0.54 | 27.57 ±2.50 | **88.09** ±0.13 | 8.52 ±0.73 |
| CFL | 42.47 ±0.02 | 32.37 ±0.09 | 67.14 ±0.87 | 26.19 ±0.26 | 52.90 ±1.48 | 1.65 ±0.96 |
| FedSoft | 91.35 ±0.04 | 19.88 ±0.50 | 83.08 ±0.02 | 22.00 ±0.50 | 85.80 ±0.29 | 1.85 ±0.21 |
| FedRC-FT | 91.02 ±0.26 | **62.37** ±1.09 | 82.81 ±0.90 | **65.33** ±1.80 | 87.14 ±0.39 | **16.27** ±0.09 |

**Ablation study on the number of concepts.**    We change the number of concepts in $[1, 3, 5]$, and report the local and global accuracy in Table 7. Results show that: 1) FedRC always achieves the best global accuracy compare with other algorithms, indicating the robustness of trained models by FedRC. 2) Although FedEM may achieve better local accuracy, the generalization ability of trained models is poor and we believe that it is an over-fitting to local data.

**Ablation studies when the number of clusters is less than the number of concepts.**    We conduct experiments with $K = 2$ and the number of concepts to 3 in Table 9, results show that FedRC still outperforms other methods.

**Ablation studies on different magnitudes of noise.**    We experimented with varying noise magnitudes, particularly focusing on larger magnitudes as shown in Table 10. The results demonstrate that: (1) FedRC can effectively handle a relatively high magnitude of noise. In our experiments, the expected value of $\xi_i$ is 45 before noise addition. We found that setting $\xi_i = 25$ ensures privacy without sacrificing performance. (2) Systems with more clients can accommodate larger levels of noise. As illustrated in Figure 14(a), adding noise up to 50 has a slight impact on performance with 300 clients, while it significantly affects the performance of FedRC with 100 clients.

**Ablation studies using shared feature extractors.**    To reduce the communication and computation costs of FedRC, we are considering using shared feature extractors among clusters. As reported in Table 11, utilizing shared feature extractors not only significantly reduces communication and computation costs but also improves the performance of FedRC.

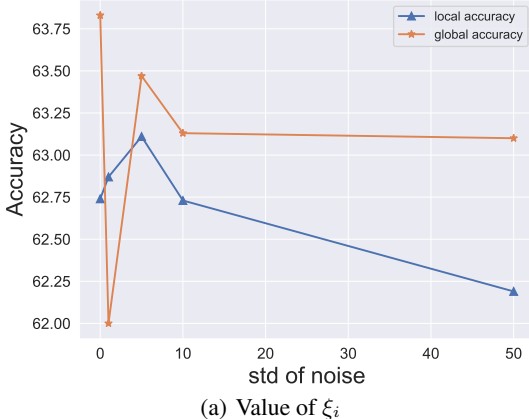

(a) Value of $\xi_i$

Figure 14: **Ablation studies on the magnitude of noise.** We shows FedRC's local and global accuracies with different noise std values $\xi_i$.

Table 5: **Ablation study on the number of local epochs.** We split CIFAR10 dataset into 300 clients, and set the number of local epochs to $\{1, 5\}$. We run algorithms for 200 communication rounds, and report the global accuracy on the round that achieves the best train accuracy.

| Algorithm | 1 | | 5 | |
|---|---|---|---|---|
| | Local | Global | Local | Global |
| FedAvg | 30.28 | 30.47 | 29.75 | 29.60 |
| IFCA | 43.76 | 26.62 | 43.49 | 35.50 |
| FeSEM | 45.32 | 30.79 | 38.32 | 24.33 |
| FedSoft | 83.08 | 22.00 | 82.20 | 19.67 |
| FedEM | 51.31 | 43.35 | 55.69 | 50.17 |
| FedRC | 62.74 | 63.83 | 57.31 | 57.43 |

**Ablation studies on scenarios with an imbalance in the number of samples across clusters.** We conducted experiments in which the number of samples in each cluster followed a ratio of 8:1:1, as shown in Table 12. Results show that FedRC maintained a significantly higher global accuracy when compared to other methods.

**Ablation studies on the number of clusters.** We conducted ablation studies on the number of clusters, as shown in Table 14. The results indicate that (1) FedRC consistently achieves the highest global accuracy across all algorithms, and (2) while increasing the number of clusters improves local accuracy in clustered FL approaches, it often has a detrimental effect on global accuracy, which is the key metric in this study.

**FedRC has the potential to handle label noise scenarios.** Following the settings in Fang & Ye (2022); Xu et al. (2022), we examined the performance of FedRC in label noise scenarios. The results show that FedRC outperforms other clustered FL methods and FedAvg in label noise scenarios.

### I.4 ABLATION STUDY ON ADAPTIVE FEDRC

In this section, we present the ablation studies on value of $\delta$ and the convergence curve of FedRC compare with FedRC-Adam.

**Ablation studies on value of $\delta$.** We vary the value of $\delta$ as the threshold for removing models, and results in Figures 15 and 16 show that: 1) both FedRC and FedRC-Adam can find the true number of concepts. 2) FedRC-Adam is more robust to $\delta$, while $\gamma_{ijk}$ in FedRC converge slower as $\delta$ decreases. 3) FedRC-Adam converges faster initially, but the final results are similar.

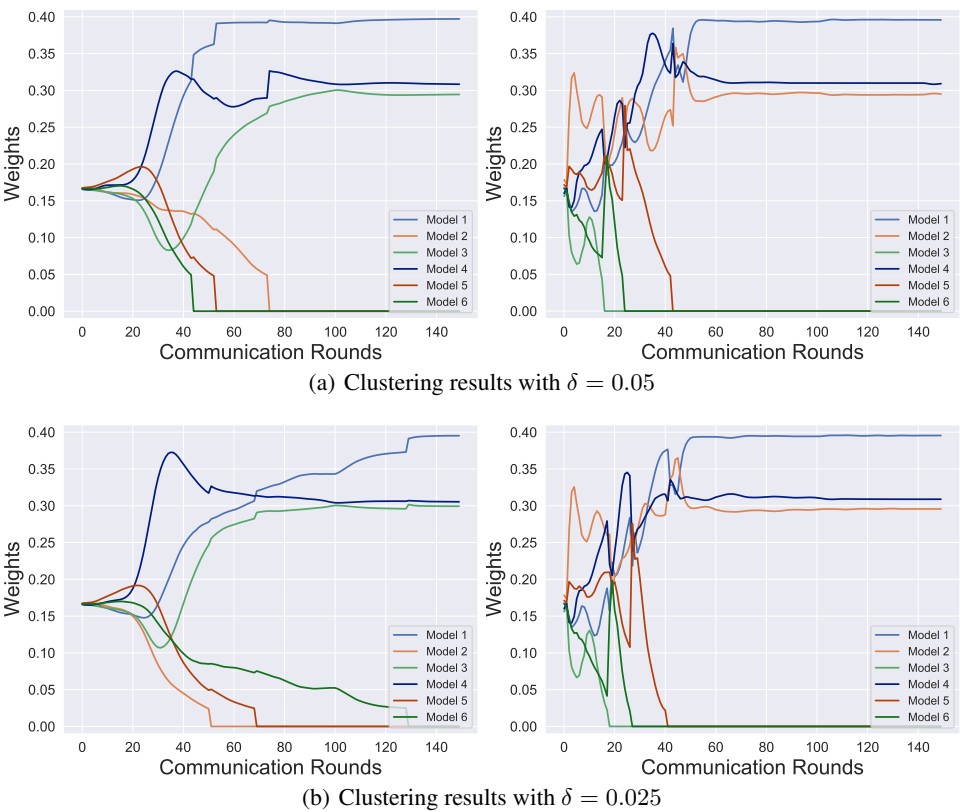

(a) Clustering results with $\delta = 0.05$

(b) Clustering results with $\delta = 0.025$

Figure 15: **Clustering results of FedRC and FedRC-Adam on different** $\delta$ We split CIFAR10 dataset to 300 clients, initialize 6 models, and report clustering results by calculating weights by $\sum_{i,j} \gamma_{i,j;k} / \sum_{i,j,k} \gamma_{i,j;k}$, which represents the portion of clients that choose the model $k$.

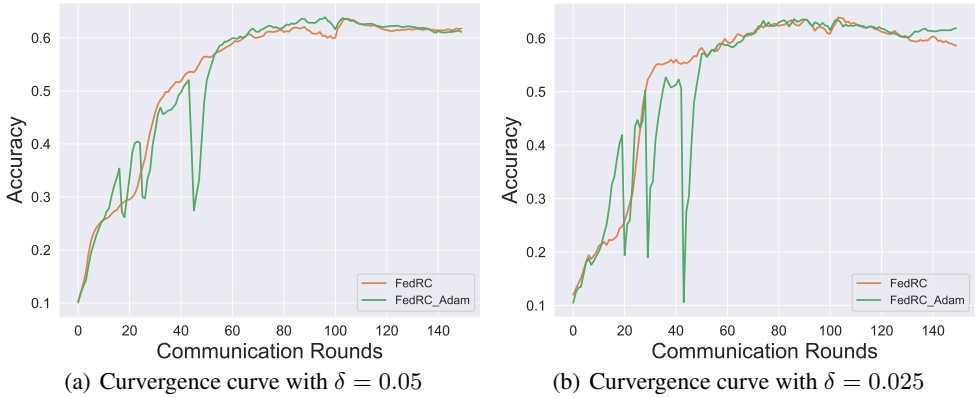

(a) Curvergence curve with $\delta = 0.05$        (b) Curvergence curve with $\delta = 0.025$

Figure 16: **Convergence curve of FedRC and FedRC-Adam with different** $\delta$**.** We split CIFAR10 dataset to 300 clients, initialize 6 models, and report the convergence curve of FedRC and FedRC-Adam with $\delta = [0.05, 0.025]$. We use adaptive process, and models is removed to 3 as in Figure 15.

Table 6: **Ablation study on the number of clients participating in each round.** We Split CIFAR10 dataset into 300 clients, and choose $\{20\%, 40\%, 60\%, 80\%, 100\%\}$ clients in each round. We run algorithms for 200 communication rounds, and report the global accuracy on the round that achieves the best train accuracy.

| Algorithm | 0.2 | | 0.4 | | 0.6 | | 0.8 | | 1.0 | |
|---|---|---|---|---|---|---|---|---|---|---|
| | Local | Global | Local | Global | Local | Global | Local | Global | Local | Global |
| FedAvg | 29.22 | 31.10 | 29.61 | 29.43 | 30.63 | 31.17 | 30.46 | 29.03 | 30.28 | 30.47 |
| IFCA | 31.95 | 16.67 | 38.07 | 23.33 | 41.26 | 29.57 | 58.86 | 49.50 | 43.76 | 26.62 |
| FeSEM | 29.86 | 25.37 | 38.31 | 28.00 | 36.05 | 32.93 | 40.53 | 27.53 | 45.32 | 30.79 |
| FedSoft | 66.59 | 10.03 | 67.37 | 9.20 | 70.76 | 10.00 | 85.37 | 10.20 | 83.08 | 22.00 |
| FedEM | 52.65 | 36.17 | 51.75 | 32.23 | 52.81 | 44.87 | 52.44 | 47.17 | 51.31 | 43.35 |
| FedRC | 61.33 | 62.10 | 62.02 | 64.2 | 65.24 | 65.07 | 63.97 | 63.90 | 62.74 | 63.83 |

Table 7: **Ablation study on the number of concepts.** We split CIFAR10 dataset to 300 clients, and change the number of concepts to $[1, 3, 5]$, and initialize $[3, 3, 5]$ models respectively. We report the local and global accuracy on the round that achieves the best train accuracy for each algorithm.

| Algorithm | 1 | | 3 | | 5 | |
|---|---|---|---|---|---|---|
| | Local | Global | Local | Global | Local | Global |
| FedAvg | 53.63 | 61.90 | 30.28 | 30.47 | 18.55 | 16.54 |
| FeSEM | 53.25 | 48.00 | 45.32 | 30.79 | 26.84 | 17.96 |
| FedEM | 64.37 | 58.00 | 51.31 | 43.35 | 51.82 | 17.76 |
| FedRC | 58.78 | 64.20 | 62.74 | 63.83 | 39.27 | 39.34 |

Table 8: **Ablation study on single-type distribution shift scenarios** We split CIFAR10 dataset into 100 clients, and set the number of local epochs to 1. We run algorithms for 200 communication rounds, and report the global accuracy and local accuracy on the round that achieves the best train performance.

| Algorithm | Feature Shift Only | | Concept Shift Only | | Label Shift Only | |
|---|---|---|---|---|---|---|
| | Local | Global | Local | Global | Local | Global |
| FedAvg | 70.92 | 79.30 | 33.76 | 33.00 | 65.88 | 65.80 |
| IFCA | 71.68 | 80.90 | 77.68 | 77.87 | 36.68 | 14.80 |
| FeSEM | 66.78 | 77.10 | 40.18 | 38.97 | 50.52 | 26.80 |
| FedEM | 69.28 | 80.00 | 77.60 | 78.57 | 78.04 | 70.00 |
| FedRC | 68.94 | 81.40 | 77.22 | 78.00 | 70.70 | 70.60 |

Table 9: **Performance of algorithms with $K = 2$.** We evaluated the performance of algorithms with insufficient number of clusters. We initialized 2 clusters for clustered FL methods and reported mean local and global test accuracy on the round that achieved the best train accuracy for each algorithm. The CIFAR10 dataset is split to 100 clients and has 3 concepts.

| Algorithms | FedAvg | FeSEM | FedEM | FedRC |
|---|---|---|---|---|
| Local Acc | 28.74 | 30.50 | 42.48 | 43.82 |
| Global Acc | 28.43 | 21.03 | 30.57 | 42.50 |

## I.5 ABLATION STUDIES ON SINGLE-TYPE DISTRIBUTION SHIFT SCENARIOS.

In this section, we evaluate the performance of clustered FL algorithms on scenarios that only have one type of distribution shifts as in traditional FL scenarios. As shown in Table 8, we can find that FedRC achieve comparable performance with other clustered FL methods on these less complex scenarios.

Table 10: **Performance of FedRC with various magnitude of the noise.** We evaluated the performance of FedRC using CIFAR10 dataset with 100 clients, and vary the magnitude of the noise from 0 to 100. We initialized 3 clusters for FedRC and reported mean local and global test accuracy on the round that achieved the best train accuracy for each algorithm.

| Magnitude of the noise | $\xi_i = 0$ | $\xi_i = 10$ | $\xi_i = 25$ | $\xi_i = 50$ | $\xi_i = 100$ |
|---|---|---|---|---|---|
| Local Acc | 48.70 | 44.72 | 45.20 | 42.60 | 41.52 |
| Global Acc | 44.37 | 46.23 | 47.10 | 36.97 | 28.23 |

Table 11: **Performance of FedRC using shared feature extractors.** We evaluated the performance of FedRC using CIFAR10 dataset with 100 clients using shared feature extractors for all the clusters to mitigate the communication and computation overhead. We initialized 3 clusters for FedRC and reported global test accuracy on the round that achieved the best train accuracy for each algorithm. We also report the size of parameters we transmitted and trained in each communication round as the **Size of Parameters**.

| Algorithm | CIFAR10 | | Tiny-ImageNet | |
|---|---|---|---|---|
| | Size of Parameters | Global Acc | Size of Parameters | Global Acc |
| FedRC | 6.71M | 44.37 | 7.44M | 28.47 |
| FedRC (shared feature extractor) | 2.26M | 54.53 | 2.99M | 33.80 |

Table 12: **Performance of algorithms with imbalance in the number of samples across clusters.** We evaluated the performance of algorithms with imbalance in the number of samples across clusters. In detail, we split CIFAR10 into 100 clients, and each concept has [80, 10, 10] clients. We initialized 3 clusters for clustered FL methods and reported mean local and global test accuracy on the round that achieved the best train accuracy for each algorithm.

| Algorithms | FeSEM | IFCA | FedEM | FedRC |
|---|---|---|---|---|
| Local Acc | 38.20 | 39.72 | 55.96 | 55.64 |
| Global Acc | 16.70 | 16.37 | 26.30 | 45.77 |

Table 13: **Performance of algorithms in label noise scenarios.** We evaluated the performance of algorithms in the label noise scenarios. In detail, we split CIFAR10 into 100 clients, and Pairflip is to randomly convert the labels. Symflip is to change $y$ to $(y + 1)\%10$. $\sigma$ is the noisy rate. We initialized 3 clusters for clustered FL methods and reported the global test accuracy on the round that achieved the best train accuracy for each algorithm. The CIFAR10 dataset is split to 100 clients.

| Algorithms | FedAvg | FeSEM | IFCA | FedEM | FedRC |
|---|---|---|---|---|---|
| Pairflip, $\sigma = 0.2$ | 52.35 | 35.25 | 20.55 | 57.55 | 59.95 |
| Symflip, $\sigma = 0.2$ | 52.60 | 32.40 | 30.35 | 53.00 | 55.25 |

Table 14: **Performance of algorithms using different number of clusters.** We evaluated the performance of algorithms using CIFAR10 and Tiny-ImageNet datasets with 100 clients. We initialized 3 and 5 clusters for clustered FL algorithms and reported local and global test accuracy on the round that achieved the best train accuracy for each algorithm.

| Algorithm | IFCA | | FeSEM | | FedEM | | FedRC | |
|---|---|---|---|---|---|---|---|---|
| | Local Acc | Global Acc | Local Acc | Global Acc | Local Acc | Global Acc | Local Acc | Global Acc |
| CIFAR10 (3 cluster) | 41.12 | 16.53 | 31.74 | 14.80 | 49.18 | 26.00 | 48.70 | 44.37 |
| CIFAR10 (5 cluster) | 45.42 | 11.07 | 41.26 | 12.63 | 61.14 | 27.27 | 51.48 | 46.67 |
| Tiny-ImageNet (3 cluster) | 23.34 | 13.57 | 23.59 | 11.93 | 27.51 | 15.83 | 34.48 | 28.47 |
| Tiny-ImageNet (5 cluster) | 28.12 | 11.03 | 28.68 | 11.03 | 31.92 | 17.77 | 38.18 | 28.07 |

# J DISCUSSION: DEFINITION OF CONCEPT SHIFTS

Following the definitions outlined in the dataset shift literature, specifically Definition 4 in Moreno-Torres et al. (2012) and Source 2 in Lu et al. (2018), we categorize concept shifts into the following classifications:

Table 15: **Performance of FedRC using hard clustering for optimization and prediction.** We split CIFAR10 and Tiny-ImageNet datasets to 100 clients, and report the global test accuracy on the round that achieved the best train accuracy for each algorithm. The (original) FedRC is trained using soft clustering, and the predictions of all models are also ensembled to derive the final prediction results. FedRC + TeHC is trained using soft clustering but only employs the models with the highest clustering weights for prediction. FedRC + TrHC is both trained and tested using a hard clustering method. This approach optimizes only the models with the highest clustering weights in local optimization steps and also uses a single model for prediction.

| Algorithms | FedRC | FedRC + TeHC | FedRC + TrHC |
|---|---|---|---|
| CIFAR10 | 44.37 | 48.03 | 20.5 |
| Tiny-ImageNet | 27.79 | 29.23 | 11.47 |

- Instances where $p_i(y|x) \neq p_j(y|x)$ and $p_i(x) = p_j(x)$ in $X \to Y$ problems, indicating "same feature different labels".
- Instances where $p_i(x|y) \neq p_j(x|y)$ and $p_i(y) = p_j(y)$ in $Y \to X$ problems, signifying "same label different features".

Here, $X \to Y$ denotes the utilization of $X$ as inputs to predict $Y$. Following most studies in FL, this paper focuses on the $X \to Y$ problems, thereby investigating "same feature different labels" problems. Moreover, within the context of $X \to Y$ problems, we contend that "same label different features" aligns more closely with the definition of "feature shift" rather than concept shift, as depicted in Clients 2 and 3 of Figure 1. Notably:

- Feature shift scenarios, such as those involving augmentation methods (CIFAR10-C, CIFAR100-C) employed in our paper, or natural shifts in domain generalization, often give rise to "same label different features" issues.
- In "same label different features" scenarios, where the same $x$ is not mapped to different $y$ values, shared decision boundaries persist, obviating the need for assignment into distinct clusters.

Consequently, we treat the challenge of "same feature different labels" as a manifestation of "feature shift" in this paper.

