# OpenReview forum: "FedRC: Tackling Diverse Distribution Shifts Challenge in Federated Learning by Robust Clustering"
_ICLR.cc/2024/Conference — Submitted to ICLR 2024_

### Official Review · Reviewer_gsbU · 2023-10-24

**Soundness:** 2 fair
**Presentation:** 3 good
**Contribution:** 3 good
**Rating:** 5
**Confidence:** 4

**Summary:**

This paper studied the federated learning problem with heterogeneous data induced by multiple types of distribution shifts, e.g., feature shift, label shift, and concept shift. To solve this problem, this paper introduced a principle of robust clustering where clients with concept shifts should be clustered together. Then, it proposed a novel FedRC approach (as well as its centralized version RobustCluster) to find the clusters based on the types of distribution shifts. The convergence of RobustCluster was also theoretically analyzed.

**Strengths:**

**Originality:** This paper studied a more challenging clustered FL setting where multiple types of distribution shifts existed in local clients. It pointed out that the clusters with concept shifts might not learn a common decision boundary, and existing clustered FL approaches failed in handling the concept shifts. This paper then proposed a novel FedRC with concept shift aware objective function. Experiments demonstrated that FedRC achieved better performance than clustered FL baselines in various data sets.

**Quality:** The motivating example in Figures 3&4 clearly illustrated the principles of robust clustering in handling concept shifts. Then the principles of robust clustering also guided the design of the objective function in Eq. (1). The clients with concept shifts were expected to clustered in different groups. The experiments verified the effectiveness of FedRC with respect to local and global generalization performance.

**Clarity:** The motivation of this paper is clear. Different from feature or label shift, concept shift essentially affects the clustering structure. The objective function in Eq. (1) aims to avoid generating clusters with concept shifts. Experiments show that the proposed FedRC significantly outperforms existing clustered FL methods.

**Significance:** The problem studied in this paper is practical but challenging. In real scenarios, different types of distribution shifts occur simultaneously among clients. As a result, adaptively generating clusters based on the types of distribution shifts can be applied to solve real-world federated learning problems.

**Weaknesses:**

**W1:** The impact of feature and label shifts on clustering can be further explained. The goal of the proposed clustered method is to separate clients with concept shifts into different clusters. It might consider clients with feature and label shifts into a single cluster. Thus, the clustering quality can also be affected by the feature and label shifts. For example, a single model might fail to hand clients with large label shifts.

**W2:** The optimization of Eq. (2) is unclear.
- Firstly, the definition of $\tilde{\mathcal{I}}(\mathbf{x}, y; \theta_k)$ is confusing. It is defined over the weights $\gamma_{i,j; k}$, but $\gamma_{i,j; k}$ is also defined over $\tilde{\mathcal{I}}(\mathbf{x}, y; \theta_k)$ in Eq. (4).
- Secondly, the updating in Eqs. (4)(5) are not associated with $\lambda_i$ in Eq. (2). Then how would the second term of Eq. (2) affect the optimization?

**W3:** The convergence of FedRC is not provided. Theorem 4.3 shows the convergence of the centralized version of FedRC. Can it also hold for federated learning scenarios?

**W4:** Step 2 in Algorithm 1 is not explained. It is unclear why checking and removing models are necessary for FedRC during training.

**Questions:**

**Q1:** Figure 1 is hard to follow. It is confusing how label shift and concept shift are involved in Figure 1.

**Q2:** Section 3 compares different clustered FL algorithms in Figure 3. It shows that existing approaches, e.g., FeSEM, IFCA, are not robust to feature and label shifts. But it is confusing how these observations are indicated in Figure 3.

**Q3:** Does FedRC in Algorithm 1 update $\gamma_{i,j; k}, w_{i; k}$ once and $\theta_{i; k}$ for $\Gamma$ local steps?

**Q4:** How are the models of nonparticipating clients generated during testing in the experiments?

**Q5:** Figure 6(d) shows that FedRC outperforms FedAvg and retains robustness when there is only one concept. When there is only one concept, would FedRC exactly recover FedAvg?

**Q6:** Figure 6(c) shows that FedRC with hard clustering consistently outperforms that with soft clustering. Besides, hard clustering can better satisfy the principles of robust clustering by separating clients with concept shifts into different clusters. In this case, It is confusing why not simply apply hard clustering when optimizing FedRC.

---

> ### Author Response · Authors · 2023-11-14
> **Response to Reviewer gsbU (1/3)**
>
> Thank you for dedicating your time to review our work. We appreciate your thoughtful feedback and insightful comments, which have undoubtedly enriched the quality of our submission. Below, we address the major concerns you raised:
>
> ### W1. The impact of feature and label shifts on clustering can be further explained. The goal of the proposed clustered method is to separate clients with concept shifts into different clusters. It might consider clients with feature and label shifts into a single cluster. Thus, the clustering quality can also be affected by the feature and label shifts. For example, a single model might fail to hand clients with large label shifts.
>
> Thank you for your valuable suggestion. We concur with the reviewer's observation that a single model may encounter challenges in handling clients with substantial label shifts. To tackle this issue, we recommend incorporating FedRC with other methods:
>
> - **Integration with algorithms tailored for label shifts:** As depicted in Figure 2 (c), amalgamating FedRC with techniques specifically designed to handle label shifts, such as FedProx and FedDecorr, significantly amplifies performance.
> - **Integration with PFL methods:** As evidenced in Table 4, FedRC can serve as a robust initialization for PFL methods, achieving performance comparable to other PFL methods through fine-tuning with just one local epoch.
>
> Furthermore, we believe that retaining models trained with FedRC, showcasing commendable generalization performance, remains invaluable. For instance, these models can be utilized by non-participating clients without the necessity for further training on model parameters.
>
> ### W2 (1). The definition of $\tilde{\mathcal{I}}(x,y;\theta_k)$ is confusing. It is defined over the weights $\gamma_{i,j;k}$ , but $\gamma_{i,j;k}$ is also defined over $\tilde{\mathcal{I}}(x,y;\theta_k)$ in Eq. (4).
>
> We apologize for any confusion, and we have made enhancements to the revised paper to ensure greater clarity:
>
> - Equation (3) *defines* $\tilde{\mathcal{I}}(x,y;\theta_k)$  by incorporating the sample-level clustering weights $\gamma_{i, j; k}$.
> - Equation (4) illustrates the *updating process* of $\gamma_{i, j; k}^{t}$ using information from previous rounds, such as $\tilde{\mathcal{I}}(x, y; \theta_k^{t-1})$.
>
> ### W2 (2). The updating in Eqs. (4)(5) are not associated with $\lambda_{i}$ in Eq. (2). Then how would the second term of Eq. (2) affect the optimization?
>
> Thank you for your insightful comment. We have set $\lambda = \frac{-N_i}{N}$ to ensure that $\sum_{k=1}^{K} \omega_{i;k} = 1$, where $N_i$ represents the number of samples from client $i$, and $N = \sum_{i=1}^{M} N_i$.
>
> We have improved the proof in Appendix A in the revised paper by adding the following clarification, emphasizing that the condition $\sum_{k=1}^{K} \omega_{i;k} = 1$ is guaranteed only when $\lambda = \frac{-N_i}{N}$.  We also list the key steps below.
>
> Starting from Equation (12) in Appendix A, we derive the following expression: $ \frac{1}{N} \sum_{j=1}^{N_i} \frac{\gamma_{i,j;k}}{\omega_{i;k}} = -\lambda_i$.
>
> Consequently, we obtain: $\omega_{i;k} = -\frac{1}{N \lambda_i} \sum_{j=1}^{N_i} \gamma_{i,j;k}$.
>
> By setting $\sum_{k=1}^{K} \omega_{i;k} = 1$ and considering that $\sum_{k=1}^{K} \gamma_{i,j;k} = 1$, we directly derive the result $\lambda = \frac{-N_i}{N}$.
>
> ### W3: The convergence of FedRC is not provided. Theorem 4.3 shows the convergence of the centralized version of FedRC. Can it also hold for federated learning scenarios?
>
> Thank you for your detailed comment. In response to the reviewer's comment, we would like to highlight that we have included a detailed discussion on the convergence of FedRC in Appendix C of our original submission. Our analysis demonstrates that FedRC can achieve a convergence rate of $O(1/\sqrt{T})$ in federated learning scenarios.
>
> ### W4: Step 2 in Algorithm 1 is not explained. It is unclear why checking and removing models are necessary for FedRC during training.
>
> We sincerely apologize for the oversight and appreciate your valuable feedback. We would like to provide clarification regarding the concerns raised:
>
> 1. **Explanation of Step 2 in Algorithm 1:**
>     - In our initial submission, Step 2 of Algorithm 1 uses adaptive FedRC to autonomously determine the number of clusters. The detailed explanation is available in Appendix F.
> 2. **Addressing Feedback:**
>     - In response to your feedback, we have removed Step 2 from Algorithm 1 on the main pages to streamline the presentation.
> 3. **Comprehensive Presentation in Appendix F:**
>     - To maintain the logical consistency of our paper while ensuring the completeness of the algorithmic details, we have incorporated the comprehensive version of Algorithm 1 in its entirety, including the previously omitted Step 2, in Appendix F.

---

> ### Author Response · Authors · 2023-11-14
> **Response to Reviewer gsbU (2/3)**
>
> ### Q1: Figure 1 is hard to follow. It is confusing how label shift and concept shift are involved in Figure 1.
>
> We apologize for the oversight regarding missing details. In response to your feedback, we have revised the paragraph as follows:
>
> > Label shifts are represented by clients exhibiting data points of varying colors, as seen in clients 1 and 2. Feature shifts are exemplified by clients maintaining data points with the same color but having substantial distances between them, as observed in clients 2 and 3. Concept shifts occur when data points at the same position have different labels, as evident in clients 2 and 5.
>
> Furthermore, Figure 1 has been revised to illustrate that clients 1, 2, and 3 coexist within the same coordinate space as clients 3, 4, and 5. This modification aims to enhance the clarity of the concept shifts for a more comprehensive understanding.
>
> ### Q2: Section 3 compares different clustered FL algorithms in Figure 3. It shows that existing approaches, e.g., FeSEM, IFCA, are not robust to feature and label shifts. But it is confusing how these observations are indicated in Figure 3.
>
> Thank you for your insightful comment. The depicted observations are elucidated in the 'Class Labels' and 'Feature Styles' rows of Figure 3, where circle sizes are employed to denote sample quantities.
>
> In the 'Class Labels' row, the x-axis represents 'labels', and the y-axis represents 'clusters'. The circle size at coordinates (y, k) indicates the number of samples with class label $y$ that choose cluster $k$. The findings suggest that in FeSEM, IFCA, and FedEM, samples with shared class labels tend to choose the same cluster. Consequently, when faced with data containing out-cluster labels, cluster models may demonstrate suboptimal performance.
>
> Similar patterns are evident in the 'Feature Styles' row concerning feature shifts.
>
> ### Q3: Does FedRC in Algorithm 1 update $\gamma, \omega$ once and $\theta$ for $\mathcal{T}$ local steps?
>
> In FedRC, we update the parameters $\gamma$ and $\omega$ once during each communication round, while $\theta$ undergoes an update for $\mathcal{T}$ local steps. To improve clarity, we have made revisions to the paper, indicating the changes with red lines in Algorithm 1.
>
> ### Q4: How are the models of nonparticipating clients generated during testing in the experiments?
>
> We apologize for the confusion. As illustrated in Appendix H.2, when performing tests on non-participating clients, we will utilize the global models stored by the servers directly. Non-participating clients are required to initially update $\gamma_{i,j;k}$ and $\omega_{i;k}$ using their corresponding local validation/training datasets before proceeding to the testing phase.
>
> Furthermore, as shown in Appendix H.2, the prediction results are generated by $y_{\text{pred}} = \sum_{k=1}^{K} \omega_{i;k} \sigma(m_{i,k}(x, \theta_k))$, where $y_{\text{pred}}$ is the predicted output, $m_{i,k}(x, \theta_k)$ is the output of model $\theta_k$ given data $x$, and $\sigma$ is the softmax function.
>
> ### Q5: Figure 6(d) shows that FedRC outperforms FedAvg and retains robustness when there is only one concept. When there is only one concept, would FedRC exactly recover FedAvg?
>
> We apologize for any confusion and appreciate the opportunity to address the concerns raised by the reviewer. We would like to provide clarification on the following points:
>
> - *In FedRC, the number of concepts may not necessarily align with the number of clusters.* It is important to note that we can set the cluster number $K > 1$ even in scenarios with only one concept. FedRC recovers FedAvg when $K = 1$, but by choosing $K = 3$ in Figure 6 (d), we aim to showcase the performance gain in terms of global test accuracy under non-concept shift conditions.
> - Through our careful observations, we have noted that in scenarios without concept shifts, FedRC tends to concentrate the majority of samples within the same cluster, while also exhibiting outliers in other clusters. Our hypothesis suggests that the observed performance enhancement may be attributed to the effective elimination of outliers within the datasets.

---

> ### Author Response · Authors · 2023-11-14
> **Response to Reviewer gsbU (3/3)**
>
> ### Q6: Figure 6(c) shows that FedRC with hard clustering consistently outperforms that with soft clustering. Besides, hard clustering can better satisfy the principles of robust clustering by separating clients with concept shifts into different clusters. In this case, It is confusing why not simply apply hard clustering when optimizing FedRC.
>
> We apologize for any confusion and appreciate the opportunity to provide clarification:
>
> - *FedRC with hard clustering in Figure 6 (c) employs hard clustering exclusively during testing while maintaining soft clustering during training.* The revised paper has been updated with red lines to address any ambiguities.
> - *The feasibility of test-time hard clustering arises from the gradual convergence of clustering weights $\gamma_{i,j;k}$ to $1$ during the training process.* The values of $\gamma_{i,j;k}$ at the final training epoch are accessible in the **`division-results/`** folder within the updated Supplementary Material.
> - *Optimizing FedRC with hard clustering introduces discrepancies between theory and practice, resulting in a decline in performance.* Theoretically, as outlined in Equations (4), (5), and Theorem 1 of Appendix A, to maximize the objective function defined in Equation (2), optimization of all $K$ models is necessary.
>
>     Furthermore, we conducted experiments below to demonstrate the performance degradation of FedRC when optimized with hard clustering. The experimental settings are consistent with those depicted in Figure 6 (c). Note that
>
>     - The (original) FedRC is trained using soft clustering, and the predictions of all models are also ensembled to derive the final prediction results (for further details, refer to Q4).
>     - FedRC + TeHC is trained using soft clustering but only employs the models with the highest clustering weights for prediction.
>     - FedRC + TrHC is both trained and tested using a hard clustering method. This approach optimizes only the models with the highest clustering weights in local optimization steps and also uses a single model for prediction.
> |  | FedRC | FedRC + TeHC | FedRC + TrHC |
> | --- | --- | --- | --- |
> | CIFAR10 | 44.37 | 48.03 | 20.5 |
> | Tiny-ImageNet | 27.79 | 29.23 | 11.47 |

---

> ### Author Response · Authors · 2023-11-20
> **Looking forward to your reply**
>
> Dear Reviewer gsbU
>
> We sincerely value your insightful review, as it plays a pivotal role in enhancing the quality of our manuscript. Moreover, we have carefully considered your feedback to further refine our paper. To review a summary of these revisions, please visit the [Summary of Revision](https://openreview.net/forum?id=6FAH0SgQzO&noteId=NN4i1FyAoB).
>
> As the discussion phase comes to a close, we kindly request that you reconsider the score if we have effectively addressed your concerns. Furthermore, we are more than willing to address any additional questions or concerns you may have.
>
> Warmest regards,
>
> Authors

---

> > ### Comment · Reviewer_gsbU · 2023-11-20
> > **Follow-up questions**
> >
> > Thanks for your response. I have some follow-up questions after reviewing the rebuttals.
> >
> > (1) The explanation on W1 is unclear to me. It mentioned that FedRC could be incorporated with other methods to handle feature and label shifts in each cluster. Does it mean that FedRC is designed for concept shift only? If not, how FedRC can handle concept/label/feature shifts in a unified framework?
> >
> > (2) Eq. (3) defines $\tilde{I}$ over $\gamma_{i,j;k}$ which represents the weight of data assigned to $\theta_k$. It might be more convincing to explain the definition of $\gamma_{i,j;k}$ in Eq. (4), e.g., how Eq. (4) is correlated with the weight of data assigned to $\theta_k$. As shown in the appendix, in Eqs. (4)(5), $w^t_{i,k}$ and $\theta_k^t$ are derived by optimizing Eq. (2). But $\gamma_{i,j;k}$ in Eq. (4) is a new definition without much explanation.
> >
> > (3) If $\lambda_i = -N_i / N$, the derivation step from Eq. (15) to Eq. (16) in the appendix is confusing. Should $w_{i;k}$ involve the negative term in Eq. (16)?
> >
> > (4) Why is the checking and removing step necessary for the proposed FedRC method? Does the convergence of FedRC in
> > Theorem C consider the impact of the proposed FedRC method? For example, would $\mathbf{\Theta}$ include the removed model parameters during training?
> >
> > (5) Since $\gamma$ and $w$ are only updated once for each communication, the number of local epochs might balance the training efficiency and the optimality (e.g., Eqs (4)(5)) for maximizing Eq. (2). Though some empirical results are provided in Table 5, it is unclear whether the sensitivity of FedRC w.r.t. the number of local epochs is induced by the sub-optimal $\gamma$ and $w$.

---

> > > ### Author Response · Authors · 2023-11-21
> > > **Response to Follow-up Questions (1/2)**
> > >
> > > We thank the reviewer for your discussion and the time you've dedicated to reviewing our manuscript and rebuttal; we truly appreciate it. We address the new comments below:
> > >
> > > ### 1. The explanation on W1 is unclear to me. It mentioned that FedRC could be incorporated with other methods to handle feature and label shifts in each cluster. Does it mean that FedRC is designed for concept shift only? If not, how FedRC can handle concept/label/feature shifts in a unified framework?
> > >
> > > Thank you for your valuable question.  We would like to clarify that
> > >
> > > - *FedRC is not designed for concept shift only, instead, it aims to distinguish concept shifts from other shift types when concept shifts, feature shifts, and label shifts **occur simultaneously**.* As illustrated in our clustering principle, the primary goal of FedRC is to distinguish concept shifts, which alert the decision boundaries, from other shift types that do not impact decision boundaries. We believe that solving this task is non-trivial and crucial for dealing with scenarios involving the concurrence of multiple shift types.
> > > - *Optimization within each cluster of FedRC follows standard FL procedures and can be integrated with other FL methods.* Once FedRC has successfully assigned clients with concept shifts into different clusters, the clients within the same clusters will only experience label and feature shifts with respect to each other, similar to traditional FL scenarios. Consequently, we can directly apply well-established methods that address feature and label shifts to enhance the optimization process of each cluster.
> > >
> > > ### 2. Eq. (3) defines $\tilde{\mathcal{I}}$ over $\gamma_{i,j;k}$, which represents the weight of data assigned to $\theta_k$. It might be more convincing to explain the definition of $\gamma_{i,j;k}$ in Eq. (4), e.g., how Eq. (4) is correlated with the weight of data assigned to $\theta_k$. As shown in the appendix, in Eqs. (4)(5), $\omega_{i;k}$ and $\theta_k$ are derived by optimizing Eq. (2). But $\gamma_{i,j;k}$ in Eq. (4) is a new definition without much explanation.
> > >
> > > We apologize for the misunderstanding in the previous response.  We would like to clarify that
> > >
> > > - As derived in Eq. (19), we have $\frac{\partial L(\Theta,\Omega)}{\partial \theta_k} = -\frac{1}{N} \sum_{i=1}^{M} \sum_{j=1}^{N_i} \gamma_{i,j;k} \nabla_{\theta_k} f(x_{i,j},y_{i,j};\theta_k)$, then we can find that $\gamma_{i,j;k}$ is the aggregation weight of each sample $(x_{i,j}, y_{i,j})$ when updating the $\theta_k$. As a result, the $\theta_k$ will fit the distribution of the samples with higher $\gamma_{i,j;k}$. Therefore, the $\gamma_{i,j;k}$ defined by the Eq. (4) is correlated with the weight of data assigned to $\theta_k$.
> > >
> > > We have further improved our paper to enhance the clarity of the definition.
> > >
> > > Revised Eq. (3):
> > >
> > > > $\tilde{\mathcal{I}} (x, y; \theta_k)=  \frac{exp (-f(x, y, \theta_k)) }{C_{y,k}}$
> > > >
> > >
> > > Revised explanation of Eq. (3):
> > >
> > > > $C_{y,k}$ is calculated using $ \frac{1}{N} \sum_{i=1}^{M} \sum_{j=1}^{N_i} 1_{\{y_{i,j}=y\}} \gamma_{i, j; k} / \frac{1}{N} \sum_{i=1}^{M} \sum_{j=1}^{N_i} \gamma_{i, j; k}$ in this paper, where  $\gamma_{i, j; k}$ represents the weight of data $(x_{i,j}, y_{i,j})$ assigned to $\theta_k$ (c.f. Remark 4.1).
> > > Thus, $C_{y,k}$ corresponds to the proportion of data pairs labeled as $y$ that choose model $\theta_k$, and can be used to approximate $P (y; \theta_k)$.
> > > >
> > >
> > > ### 3. If $\lambda = -\frac{N_i}{N}$, the derivation step from Eq. (15) to Eq. (16) in the appendix is confusing. Should $\omega_{i;k}$ involve the negative term in Eq. (16)?
> > >
> > > We apologize for the mistake. Upon review, we discovered that a '$-$' symbol was missed in Eq (14) and (15) in the appendix of the revised manuscript. This issue has been resolved, and we have updated the manuscript. The corrected versions are also available in [W2 (2) of previous response at 14 Nov](https://openreview.net/forum?id=6FAH0SgQzO&noteId=GR7ObXy5xc).

---

> ### Author Response · Authors · 2023-11-21
> **Response to Follow-up Questions (2/2)**
>
> ### 4. Why is the checking and removing step necessary for the proposed FedRC method? Does the convergence of FedRC in Theorem C consider the impact of the proposed FedRC method? For example, would $\Theta$ include the removed model parameters during training?
>
> Thank you for bringing this to our attention. We appreciate the opportunity to clarify a few key points:
>
> - **Checking and removing steps are optional:** It's important to note that the checking and removing steps are not necessary. We have explicitly removed these steps in Algorithm 1 in the main paper. In Appendix I and F, we delve into practical enhancements for FedRC, exploring techniques such as the checking and removing step, privacy-preserving methods, and Adam-like clustering weights updating. However, we want to emphasize that these techniques are not obligatory for FedRC.
> - **The theory part considers the original FedRC:** The convergence analysis provided in the theory section addresses the original FedRC, specifically referring to Algorithm 1. The techniques explored in Appendix I and F, including the aforementioned checking and removing step, are not considered in the theory part. We agree with the reviewer that it would be an interesting future direction to give a more comprehensive theoretical understanding when adding all the practical techniques together.
>
> ### 5. Since $\gamma$ and $\omega$ are only updated once for each communication, the number of local epochs might balance the training efficiency and the optimality (e.g., Eqs (4)(5)) for maximizing Eq. (2). Though some empirical results are provided in Table 5, it is unclear whether the sensitivity of FedRC w.r.t. the number of local epochs is induced by the sub-optimal $\gamma$ and $\omega$
>
> Thank you for your insightful comments and suggestions. In response to your query about the sensitivity of FedRC to suboptimal values of $\gamma$ and $\omega$, we would like to demonstrate that the experiments in Table 5 show that FedRC performs relatively robustly under variations in the number of local epochs. Furthermore, we would like to provide some theoretical insights here.
>
> Theoretically, the sub-optimal $\gamma$ and $\omega$ may hinder the convergence at the initial stages of the training:
>
> - **Preliminaries on the impact of the number of local epochs and the update of clustering weights on the convergence of FedRC.** In Appendix C, we demonstrated the construction of the surrogate function $g_i$ satisfying all the requirements outlined in [1]. Utilizing this surrogate optimization framework, we can directly derive the convergence analysis.
>
>     By extending Lemma G.1 from [1], the single-round convergence $\mathbb{E} [\mathcal{L}(\Theta^{t+1},\Omega^{t+1}) - \mathcal{L}(\Theta^{t},\Omega^{t})]$ can be lower-bounded by the sum of
>
>     1. the KL divergence between clustering weights $\frac{1}{N} \sum_{i=1}^{M} \sum_{j=1}^{N_i} KL (\gamma_{i,j;k}^{t+1} \|\| \gamma_{i,j;k}^{t})$,
>     2. the gradient norms of $\Theta^{t}$, denoted by $\frac{\mathcal{T}}{4} \mathbb{E} [ \|\| \nabla_{\Theta} L(\Theta^{t}, \Omega^{t}) \|\|^2]$,
>     3. certain negative constants measuring client drifts and related to $\mathcal{T}$.
> - **The local epochs hinder the convergence of FedRC due to client drifts.** To expedite convergence, we aim to maximize the single-round convergence $\mathbb{E} [\mathcal{L}(\Theta^{t+1},\Omega^{t+1}) - \mathcal{L}(\Theta^{t},\Omega^{t})]$. Consequently, we observe that the terms related to $\mathcal{T}$ that *slow down convergence are the constant terms measuring client drifts, not related to $\gamma$*.
> - **Negative impacts of sub-optimal $\gamma$ and $\omega$ reduce after the initial stages of training.** According to Theorem C.1, the clustering weights $\omega_{i;k} = \frac{1}{N_i} \sum_{j=1}^{N_i} \gamma_{i,j;k}$ converge faster than the parameters $\theta_k$. Thus, during the initial stages of training, when $\gamma_{i,j;k}^{t+1}$ and $\gamma_{i,j;k}^{t}$ vary significantly, the induced error will be large. However, beyond the initial stages of training, the clustering weights tend to converge, mitigating the induced error attributable to sub-optimal $\gamma$ and $\omega$.
>
> [1] Marfoq, Othmane, et al. "Federated multi-task learning under a mixture of distributions." *Advances in Neural Information Processing Systems* 34 (2021): 15434-15447.

---

> > ### Comment · Reviewer_gsbU · 2023-11-21
> > **Comments**
> >
> > (1) If I understand correctly, the proposed FedRC method can perform clustering based on concept shift, no matter whether other types (feature/label) of distribution shifts exist or not. It is a generic strategy to handle concept shifts, and it can identify concept shifts even when other distribution shifts exist among clients. But FedRC cannot directly handle the feature/label shifts in each cluster. Please correct me if there is any misunderstanding.
> >
> > (2) Did all the experiments in the paper use Algorithm 1 without the checking and removing step? Is there any result to show the impact of this step on the proposed algorithm? In addition, it seems that in the submitted code, lines 285-288 in "run_experiments.py" involve a "remove_learner" step. Does this correspond to the checking and removing step?

---

> > > ### Author Response · Authors · 2023-11-22
> > > **Response to comments**
> > >
> > > Thank you very much for your detailed comments. Please find our answers to your raised questions below.
> > >
> > > ### Q1. If I understand correctly, the proposed FedRC method can perform clustering based on concept shift, no matter whether other types (feature/label) of distribution shifts exist or not. It is a generic strategy to handle concept shifts, and it can identify concept shifts even when other distribution shifts exist among clients. But FedRC cannot directly handle the feature/label shifts in each cluster. Please correct me if there is any misunderstanding.
> > >
> > > You are correct. The FedRC can perform clustering based on concept shift, and the performance gain of FedRC over other methods comes from its ability to identify concept shifts even when other distribution shifts exist among clients.
> > >
> > > ### Q2. Did all the experiments in the paper use Algorithm 1 without the checking and removing step? Is there any result to show the impact of this step on the proposed algorithm? In addition, it seems that in the submitted code, lines 285-288 in "run_experiments.py" involve a "remove_learner" step. Does this correspond to the checking and removing step?
> > >
> > > We would like to clarify that
> > >
> > > 1. All the experiments in the main pages use Algorithm 1 without the 'checking and removing step.’
> > > 2. We present experiments demonstrating the impact of the 'checking and removing step' in Appendix I.4 of the revised manuscript (Appendix I.3 of the original submission). The results indicate that by utilizing the 'checking and removing step,' FedRC can automatically find the number of concepts. Further analysis of the experimental results can be found in Appendix F.1.
> > > 3. Yes, lines 277-291 correspond to the ‘checking and removing step’.

---

### Official Review · Reviewer_kcSi · 2023-10-29

**Soundness:** 4 excellent
**Presentation:** 3 good
**Contribution:** 3 good
**Rating:** 8
**Confidence:** 4

**Summary:**

This paper identifies the learning challenges posed by the simultaneous occurrence of diverse distribution shifts and propose a clustering principle to overcome these challenges, i.e., separating clients with concept shifts into different clusters, while keeping clients without concept shifts in the same cluster.
The principle is further translated into a bi-level optimization problem which are provided with an efficient and convergent optimizer.
Extensive experiments demonstrate that FedRC significantly outperforms other SOTA.

**Strengths:**

1. The paper identifies an important problem 'ensuring global performance when multiple types of distribution shifts occur simultaneously among clients'. The illustration of Figure 1 clearly shows the motivation.
2. The algorithm comes with theoretic analysis including convergence proof for FedRC as well as RobustCluster.
3. The experiments are presented with sufficient details, such as ablations and experiments on real-world concept shift data. Great effort.

**Weaknesses:**

1. Section 6, it seems 'future work' has already been done by the appendix. Better find 'real' future work or change the title of the last section.
The same goes for 'Limitations'.
2. It seems that FedRC outperforms previous SOTA by a large margin. The success of FedRC seems lie in the objective funtion eq. (8). However, there is a lack of theoretic comparison between eq. (8) and the obj. func of existing methods.

**Questions:**

no

---

> ### Author Response · Authors · 2023-11-14
> **Response to Reviewer kcSi**
>
> Thank you sincerely for your positive review of our work. We also greatly appreciate your constructive comments. We have carefully considered the main concerns you raised and address them below.
>
> ### 1. Section 6, it seems 'future work' has already been done by the appendix. Better find 'real' future work or change the title of the last section. The same goes for 'Limitations'.
>
> Thank you for your valuable suggestion. We have revised Section 6 to the following paragraph:
>
> > This paper addresses the diverse distribution shift challenge in FL and proposes using clustered FL methods to tackle it. However, we found that none of the existing clustered FL methods effectively address the diverse distribution shift challenge, leading us to introduce FedRC as a solution. Furthermore, we have explored extensions in Appendix I and Appendix F, including improvements in communication and computation efficiency, automatic determination of the number of clusters, and mitigation of the personalization-generalization trade-offs. For future research, delving deeper into providing a more comprehensive theoretical understanding of the distinctions in clustering results between FedRC and other methods would be intriguing.
> >
>
> In summary, we simplified the discussion of the techniques in the Appendix and introduced a 'real' future work based on the reviewer's suggestion.
>
> ### 2. It seems that FedRC outperforms previous SOTA by a large margin. The success of FedRC seems lie in the objective function eq. (8). However, there is a lack of theoretic comparison between eq. (8) and the obj. func of existing methods.
>
> Thank you for your insightful comment. We appreciate your feedback. We would like to clarify that while Equation (8) updates clustering weights $\omega$, it does not explicitly define our objective functions. We presume that you are referring to Equation (1), which indeed defines the objective function of FedRC.
>
> We acknowledge that providing an in-depth theoretical analysis of the distinctions between the clustering results obtained from Equation (1) and existing methods is challenging. However, we can delve into a discussion on the design of the objective function.
>
> Our paper includes an illustrative example to highlight the advantages of Eq (1). In Figure 9 of Appendix G, we compare RobustClustering with the traditional EM algorithm that maximizes the log-likelihood functions. The results in Figure 9 indicates:
>
> 1. **Limitation of the EM Algorithm:**
>     - Confronted with inputs featuring concept shifts and a poor initialization of $p(x, y; \theta_k)$, the EM algorithm tends to converge to suboptimal local optima. In these cases, the samples with concept shifts often get erroneously assigned to the same cluster, resulting in subpar classification outcomes.
> 2. **Advantages of RobustClustering:**
>     - In contrast, the RobustClustering algorithm showcases its ability to automatically adjust cluster assignments. This adaptive nature allows it to navigate away from unfavorable local optimums, leading to superior classification results for all samples, even in the presence of concept shifts.
>
> These observations underscore the robustness and effectiveness of our proposed RobustClustering method, especially in scenarios where traditional EM algorithms may struggle due to poor initialization and concept shifts, as illustrated in Figure 9 and experiment results in Tables 1 and 2.

---

### Official Review · Reviewer_xiXR · 2023-10-31

**Soundness:** 4 excellent
**Presentation:** 4 excellent
**Contribution:** 3 good
**Rating:** 8
**Confidence:** 4

**Summary:**

This paper proposes FedRC, a novel algorithm framework based on soft clustering, to ensure global model performance when multiple types of distribution shifts occur in clients' data, including feature shift, label shift, and concept shift. Specifically, FedRC addresses the challenges posed by distribution drift by combining the proposed clustering principles with a dual level optimization problem and a new objective function. The main contributions of this paper are:
1) This paper proposes the principle of robust clustering to address the challenges posed by multiple data distribution drift.
2) This paper proposes that FedRC implement robust clustering principles and provides theoretical analysis.
3) This paper conducts experiments on multiple datasets to demonstrate the effectiveness of the proposed method and FedRC can be integrated with existing methods.

**Strengths:**

1)	The paper is technically well presented.
2)	The proposed method is well-motivated and novelty, the analysis of the related work is clear and convincing.
3)	The paper provides rigorous theoretical analysis of the proposed method.
4)	The authors do a lot of experiments to prove that their method is good and compare it with many existing methods, the results seem convincing.

**Weaknesses:**

1) It will be clearer if there is a workflow diagram to explain the working principle of the proposed method.
2) According to the robust clustering principle proposed by the author, clients with concept drift are classified into different categories. It will be more convincing if the division results are presented.
3) In Tables 1, 2, and 4, some methods have very low accuracy on CIFAR100 or Tiny-ImageNet datasets. Authors should provide reasons for the very low accuracy in experimental analysis.
4) The writing of some symbols should be unified. In Eq. (1), $x_{ij}$ and $y_{ij}$ should be $x_{I,j}$, and $y_{I,j}$.

**Questions:**

1) Concept shift has two cases: “same label different features” and “same feature different labels”, Can the FedRC ignore the difference between the two concept shifts in this paper?
2) Figure 2 (a) is not mentioned in the main text. And according to the results in Figure 2 (a), there is concept drift in scenarios with significant improvement in FedRC. Is FedRC only more effective for concept drift?
3) In section 4.1, the authors claim that If (x, y) exhibits the concept shift with respect to the distribution of cluster k, P (y | x; θ_k) will be small. Please give a detailed explanation.
4) In Algorithm 1, does each client need to calculate local update for each clustering model? If so, should loops be added to k models in local update?
5) The title of Figure 5 mentions' Both groups have IID training and test datasets'. Does 'IID' here refer to the overall data distribution of all clients or the data distribution of each client? If the data distribution of the clients is IID, does it conflict with the settings of the participating clients?

---

> ### Author Response · Authors · 2023-11-14
> **Response to Reviewer xiXR (1/3)**
>
> We sincerely appreciate your positive review and are grateful for the time and effort you dedicated to providing valuable feedback. Please find our responses to the major concerns below.
>
> ### 1. It will be clearer if there is a workflow diagram to explain the working principle of the proposed method.
>
> Thank you for your suggestion! As per your recommendation, we have incorporated Figure 10 into the revised version of the paper to provide the workflow diagram illustrating the whole optimization process of FedRC.
>
> ### 2. According to the robust clustering principle proposed by the author, clients with concept drift are classified into different categories. It will be more convincing if the division results are presented.
>
> Thank you for your valuable suggestion. In response, we have incorporated the division results of FedRC in various formats in our revised submission.
>
> 1. **Visualization about the division results in Appendix I.2:** We have introduced a visual representation in Appendix I.2 using circles of varying sizes to depict the number of samples belonging to each cluster. The division results of FedRC on CIFAR10, CIFAR100, and Tiny-Imagenet datasets are presented in this format. Our results consistently demonstrate the successful achievement of FedRC in assigning data samples with concept shifts to different clusters.
> 2. **Raw Files on Division Results:** Additionally, we have incorporated raw files that contain the distribution of clustering weights for each sample at the end of training in the updated Supplementary Material. These files are now organized within the **`division-results/`** folder. Each file follows a structure with $N$ rows, where $N$ denotes the count of training samples. Within each row, the values represent (concept_id, label_id, feature_style_id, clustering weights). It is important to note that rows with concept_id pairs (1 and 2), (3 and 5), and (4 and 6) correspond to identical concepts.
>
> We hope these additional information provide a clearer understanding of the division results of FedRC in our paper.
>
> ### 3. In Tables 1, 2, and 4, some methods have very low accuracy on CIFAR100 or Tiny-ImageNet datasets. Authors should provide reasons for the very low accuracy in experimental analysis.
>
> Thank you for your comprehensive feedback. The notably low accuracies can be attributed primarily to the following factors:
>
> - *Clustered FL methods, with the exception of FedRC, struggle to handle concept shifts.* The presence of concept shifts within a single classifier significantly impacts the model's performance, as empirically investigated in existing literature [1].
> - *The PFL methods exhibit notably low global accuracy, primarily attributable to the simultaneous presence of concept shifts and overfitting to local distribution.*
>     - As elucidated in [2], PFL methods have a tendency to overfit local distributions, resulting in elevated local performance but diminished global performance, as demonstrated in Table 4 of our paper.
>     - In our study, global accuracy is computed by averaging across all test clients for each concept. The challenge arises from the difficulty of generalizing personalized models trained on individual concepts to other concepts, leading to a further reduction in the reported global accuracy of PFL methods.
>
> ### 4. The writing of some symbols should be unified. In Eq. (1), $x_{i,j}$ and $y_{i,j}$ should be $x_{I,j}$ and $y_{I,j}$.
>
> Thank you for your insightful suggestions! Following your guidance, we carefully reviewed the notations in Equation (1), focusing on $x_{i,j}$ and $y_{i,j}$. We observed that the current version might be correct regarding its contexts.
>
> To avoid overlooking any details, we kindly request the reviewer to provide additional information, enabling us to pinpoint and address any potential problems that may have eluded us. Your further guidance will be invaluable in refining our work.

---

> ### Author Response · Authors · 2023-11-14
> **Response to Reviewer xiXR (2/3)**
>
> ### 5. Concept shift has two cases: “same label different features” and “same feature different labels”, Can the FedRC ignore the difference between the two concept shifts in this paper?
>
> We apologize for any confusion and appreciate the opportunity to provide further clarification regarding the definition of concept shifts in our paper. Additionally, we have added Appendix J to incorporate the discussions below into our revised paper.
>
> Following the definitions outlined in the dataset shift literature, specifically Definition 4 in [3] and Source 2 in [4], we categorize concept shifts into the following classifications:
>
> 1. Instances where $p_{i}(y|x) \neq p_{j}(y|x)$ and $p_{i}(x) = p_{j}(x)$ in $X \to Y$ problems, indicating "same feature different labels."
> 2. Instances where $p_{i}(x|y) \neq p_{j}(x|y)$ and $p_{i}(y) = p_{j}(y)$ in $Y \to X$ problems, signifying "same label different features."
>
> Here, $X \to Y$ denotes the utilization of $X$ as inputs to predict $Y$. Following most studies in FL [5, 6, 7], this paper focuses on the $X \to Y$ problems, thereby investigating "same feature different labels" problems.
>
> Moreover, within the context of $X \to Y$ problems, we contend that "same label different features" aligns more closely with the definition of "feature shift" rather than concept shift, as depicted in Clients 2 and 3 of Figure 1. Notably:
>
> 1. Feature shift scenarios, such as those involving augmentation methods (CIFAR10-C, CIFAR100-C) employed in our paper or natural shifts in domain generalization, often give rise to "same label different features" issues.
> 2. In "same label different features" scenarios, where the same $x$ is not mapped to different $y$ values, shared decision boundaries persist, obviating the need for assignment into distinct clusters.
>
> Consequently, we treat the challenge of "same feature different labels" as a manifestation of "feature shift" in this paper. We hope that this clarification enhances the understanding of definitions.
>
> ### 6. Figure 2 (a) is not mentioned in the main text. And according to the results in Figure 2 (a), there is concept drift in scenarios with significant improvement in FedRC. Is FedRC only more effective for concept drift?
>
> Thank you for your feedback; we have added the reference to Figure 2 (a) in the Introduction section.
>
> The FedRC exhibits superior global accuracy, even in the absence of concept shifts. For instance, as illustrated in Table 7, FedRC showcases a notable improvement of 2.3% in global accuracy compared to alternative methods. Furthermore, the performance advantage of FedRC becomes even more pronounced in scenarios involving concept shifts.

---

> ### Author Response · Authors · 2023-11-14
> **Response to Reviewer xiXR (3/3)**
>
> ### 7. In section 4.1, the authors claim that If (x, y) exhibits the concept shift with respect to the distribution of cluster k, P (y | x; θ_k) will be small. Please give a detailed explanation.
>
> Thank you for your detailed suggestion! The assertion is grounded in the definition of concept shifts:
>
> - Concept shifts (please refer to Question 5 for the formal definition) are defined by the inequality $p_{j}(y|x) \neq p_{i}(y|x)$ for distributions $i$ and $j$. If we consider $\theta_i$ as the parameters trained on distribution $i$, the probability density of in-distribution samples $(x, y_{i})$ will inherently surpass that of out-distribution samples $(x, y_{j})$. This phenomenon is widely leveraged in research domains such as concept drift detection [4].
> - To illustrate this phenomenon, let's examine a simple example. Assume the distribution parametrized by $\theta_k$ includes data points $(x, y) = (1, 1)$ and $(x, y) = (2, 2)$. Consequently, we have $p(1|1;\theta_k) = 1.0$ and $p(2|2;\theta_k) = 1.0$. In the occurrence of concept shifts, where the same $x$ is associated with different $y$ values, as in the cases of $(1, 2)$ and $(2, 1)$, we observe that $p(2|1;\theta_k) = 0.0$ and $p(1|2;\theta_k) = 0.0$.
>
> ### 8. In Algorithm 1, does each client need to calculate local update for each clustering model? If so, should loops be added to k models in local update?
>
> Thank you for your valuable suggestion, and we apologize for the oversight in missing details. In response to your comments, we have included '$\forall k$' in line 7 of the revised manuscript.
>
> ### 9. The title of Figure 5 mentions' Both groups have IID training and test datasets'. Does 'IID' here refer to the overall data distribution of all clients or the data distribution of each client? If the data distribution of the clients is IID, does it conflict with the settings of the participating clients?
>
> We apologize for any confusion regarding the experiment settings. In the original statement, 'IID' denoted that the training and test distributions of each client are identical, but it may not have been clear that individual clients have distinct local distributions. To provide clarity, we have revised the sentence in the paper to state: 'The training and test distributions of each client are identical.
>
> [1] Shuqi Ke, Chao Huang, and Xin Liu. Quantifying the impact of label noise on federated learning. Arxiv 2022.
>
> [2] Shanshan Wu, Tian Li, Zachary Charles, Yu Xiao, Ziyu Liu, Zheng Xu, and Virginia Smith. Motley: Benchmarking heterogeneity and personalization in federated learning. Arxiv 2022.
>
> [3] Moreno-Torres J G, Raeder T, Alaiz-Rodríguez R, et al. A unifying view on dataset shift in classification. PR 2012.
>
> [4] Lu J, Liu A, Dong F, et al. Learning under concept drift: A review. IEEE TKDE 2018.
>
> [5] McMahan B, Moore E, Ramage D, et al. Communication-efficient learning of deep networks from decentralized data. AISTATS 2017.
>
> [6] Li T, Sahu A K, Zaheer M, et al. Federated optimization in heterogeneous networks. MLSys 2020.
>
> [7] Karimireddy S P, Kale S, Mohri M, et al. Scaffold: Stochastic controlled averaging for federated learning. ICML 2020.

---

### Author Response · Authors · 2023-11-14
**Summary of Revision**

First of all, we sincerely appreciate the reviewers for their valuable time and constructive comments. We have thoroughly considered their suggestions and diligently incorporated them into our revised manuscript, clearly marking the changes with the red lines. For the convenience of the reviewers, we outline the key modifications of the manuscript in the following summary.

1. [Reviewer gsbU] Additional explanations about label shifts, feature shifts, and concept shifts have been incorporated into Figure 1. Besides, we have revised Figure 1 to illustrate that clients 1, 2, and 3 coexist within the same space as clients 3, 4, and 5.
2. [Reviewer xiXR] Figure 10 of Appendix H now illustrates the workflow of FedRC, and we have referred to it in Section 4.2.
3. [Reviewer kcSi] In Section 6, we have simplified the discussion of the techniques explored in the appendices. Additionally, we have introduced 'real' future work as per the recommendation of the reviewer kcSi.
4. [Reviewer gsbU] We improved the proof in Appendix A by providing clarification that emphasizes the condition $\sum_{k=1}^{K} \omega_{i;k} = 1$ is guaranteed only when $\lambda = \frac{-N_i}{N}$.
5. [Reviewer xiXR] We added a new subsection, Appendix I.2, to present the visualization of division results of FedRC.
6. [Reviewer gsbU] We have included the experiments concerning the performance of FedRC, utilizing hard clustering for both optimization and prediction, in Table 15 of Appendix I.
7. [Reviewer xiXR] We added a new section, Appendix J, to include the discussions about the definition of concept shifts in our paper.
8. [Reviewer gsbU, kcSi, and xiXR] We have taken measures to enhance the clarity of our manuscript by refining certain sentences to mitigate potential misunderstandings. For instance, we clarified that FedRC is trained using soft clustering but tested using hard clustering in Figure 6(c). Additionally, we emphasized that the training and test distributions of each client are identical, while clients' local distributions are not identical.

---

### Meta-Review · Area_Chair_2sgx · 2023-12-23

**Metareview:**

This paper makes a new problem setting: a federated learning setup where there are 3 kinds of distribution shifts simultaneously happening. The paper proposes an algorithm for this setting, based on a clustering-based objective function. This objective function is reminiscent of an EM-style objective where each data point is soft-assigned to a cluster, and cluster parameters are updated

The paper then artificially creates a setting where these shifts are induced in existing (good) datasets, and show that existing methods drastically underperform their new method. A pro-forma theoretical result on convergence rate for smooth optimization is provided for this setting.

My main concerns with this paper are:
(1) the paper does not make a convincing case that the problem considered here - with such drastic and simultaneous problems - arises in any practical setting.
(2) even if we accept that the problem is important, it is not clearly explained why the proposed clustering-based loss function (1) is the right one to solve these issues. In particular the objective is pretty much just clustering; the paper just shows that training a cluster of models is better than training a single model for all the data in this setting. the method is not tailored to the problem.

Minor concerns:
(1) if the data is clean and good, and does not have these shifts, does the proposed method perform as well as existing methods ? or does it under-perform them since it tries to cluster data where it should not be clustered ?
(2) studying this in the federated setting does not really change anything, and (I feel) just obfuscates the issues that would be present even in a simpler centralized setting.
(3) how are the value of lambda's chosen ? are they updated or kept fixed ?
(4) the paper is a bit strangely written; for example, what is the (repeatedly referred to) "principles of robust clustering" ? it is not a precise or mathematical notion, and is also not a commonly used term anywhere.

**Justification For Why Not Higher Score:**

Please see my Metareview above

**Justification For Why Not Lower Score:**

N/A

---

### Decision · Program_Chairs · 2024-01-16

Reject